# HEUREKABENCH: A BENCHMARKING FRAMEWORK FOR AI CO-SCIENTIST

**Siba Smarak Panigrahi**[1*], **Jovana Videnović**[1,2*†], **Maria Brbić**[1]
[1]École Polytechnique Fédérale de Lausanne (EPFL)
[2]ETH Zurich
`brbiclab.epfl.ch/projects/heurekabench`

## ABSTRACT

LLM-based reasoning models have enabled the development of agentic systems that act as co-scientists, assisting in multi-step scientific analysis. However, evaluating these systems is challenging, as it requires realistic, end-to-end research scenarios that integrate data analysis, interpretation, and the generation of new insights from the experimental data. To address this limitation, we introduce HEUREKABENCH, a framework to create benchmarks with *exploratory, open-ended research questions* for experimental datasets. Each such question is grounded in a scientific study and its corresponding code repository, and is created using a semi-automated pipeline that leverages multiple LLMs to extract insights and generate candidate workflows, which are then verified against reported findings. We instantiate the framework in single-cell biology to obtain sc-HEUREKABENCH benchmark and use it to compare state-of-the-art single-cell agents. We further showcase the benefits of our benchmark for quantitatively analyzing current design choices in agentic systems. We find that the addition of a *critic* module can improve ill-formed responses for open-source LLM-based agents by up to 22% and close the gap with their closed-source counterparts. Overall, HEUREKABENCH sets a path toward rigorous, end-to-end evaluation of scientific agents, grounding benchmark construction in real scientific workflows.

## 1 INTRODUCTION

Post-training algorithms for large language models (LLMs) with reinforcement learning (RL) have led to rapid progress in improving their reasoning capabilities (Rafailov et al., 2023; Shao et al., 2024; Chai et al., 2025). As a result, LLM-based agents have been developed to solve complex tasks through multi-turn interactions with an external environment (Huang et al., 2022; Yao et al., 2023; Yang et al., 2024). These systems can be broadly characterized by a loop consisting of three stages: in the THINK stage, the agent needs to understand the user request and create a plan; in the ACT stage it generates an action depending on the plan; and in the OBSERVE stage it updates the context with the feedback of action execution for the next cycle.

Agentic systems offer promise for advancing scientific discovery by actively generating new hypotheses, designing and evaluating experiments, and iteratively improving through feedback (M. Bran et al., 2024; Ghareeb et al., 2025; Saeedi et al., 2025; Huang et al., 2025a). In this capacity, they could act as AI co-scientists, complementing human expertise and accelerating the scientific discovery process (Yamada et al., 2025; Gottweis et al., 2025). One scientific domain that has recently seen an influx of agents is single-cell biology (Xiao et al., 2024; Roohani et al., 2025; Alber et al., 2025; Mitchener et al., 2025; Huang et al., 2025b). A typical real-world use case in this domain involves autonomously proposing novel hypotheses, performing complex analyses of single-cell RNA-seq experimental datasets and delivering scientifically meaningful findings.

With this rapid progress of AI agents, it is crucial to design benchmarks to rigorously evaluate and further improve their potential as AI co-scientists. Such benchmarks should pose scenarios that require the agent to actively explore the dataset and generate data-driven insights rather than merely

---

*Equal contribution. Correspondence: `mbrbic@epfl.ch`
†Work done during the Summer@EPFL internship program

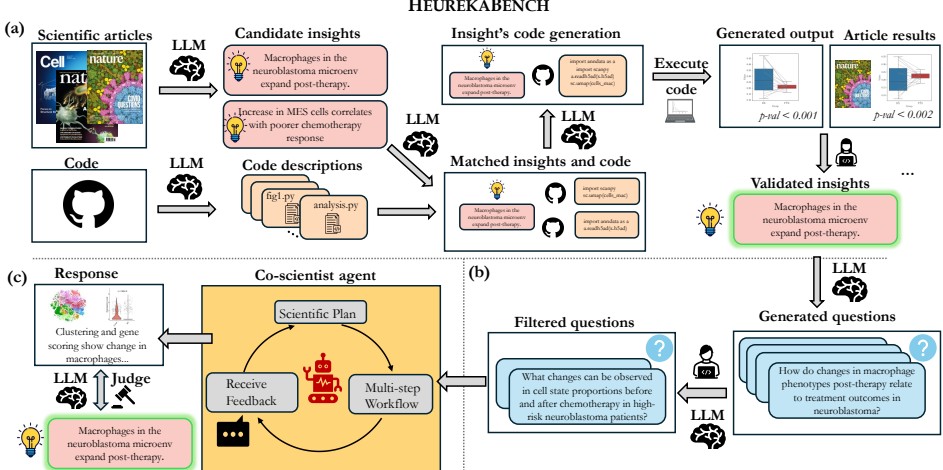

Figure 1: **Illustration of the HEUREKABENCH framework.** The HeurekaBench consists of three stages: *(a)* insight generation, where candidate insights are extracted from scientific articles and semi-automatically validated; *(b)* question generation, where validated insights are reformulated as question-answer pairs; and *(c)* question solving, where the agent autonomously designs and executes a multi-step analysis, producing a data-driven answer that is evaluated against published findings.

recalling factual knowledge. However, the current benchmarks focus on static knowledge retrieval and narrow reasoning tasks, thus failing to evaluate the open-ended nature of scientific discovery.

**Limitations of existing benchmarks.** Most existing data-driven scientific discovery benchmarks test task instruction following and answering a single computational question from experimental data (*e.g.*, *How many miRNAs remain significant at $p \leq 0.05$ after Benjamini-Hochberg correction?* from Mitchener et al. (2025) and *Train a VAE model and perform a 1-vs-all differential expression test for each cell type* from Chen et al. (2025)). However, a co-scientist should autonomously plan these questions as sub-steps within the workflow, rather than explicit user instructions. The closest to our work is BaisBench (Luo et al., 2025); however, it relies solely on the capabilities of LLMs to formulate questions from scientific studies, leading to invalidated questions that may not be possible to answer accurately. Thus, existing benchmarks lack an appropriate framework for evaluating open-ended, free-form, data-driven scientific responses that are required for an AI co-scientist.

**Our work: HEUREKABENCH.** To address these issues, we introduce HEUREKABENCH, a framework for constructing benchmarks that capture the envisioned role of co-scientists: tackling open-ended, data-driven scientific questions through hypothesis-driven exploratory analysis and iterative reasoning (Figure 1). The core idea behind the HEUREKABENCH is to *ground benchmark construction in the scientific process itself*. To achieve that, we build a multi-LLM pipeline to automatically extract key insights and match them with candidate code workflows from peer-reviewed publications and their associated code repositories. We validate insights by reproducing published results through successful code execution and comparing the generated results with the results from scientific studies, ensuring reliability and scientific grounding. From these validated insights, we derive two complementary benchmark formats: *(i)* **open-ended research questions (OEQs)**, which capture the exploratory nature of real-world co-scientist tasks and serve as the primary measure of a co-scientist performance, and *(ii)* **multiple-choice questions (MCQs)**, which provide a lightweight proxy for rapid evaluation during agent development. To rigorously evaluate open-ended agent responses, we propose a new evaluation scheme that instructs the evaluator to decompose both the agent and the ground-truth answers into *atomic facts*, and then compare the presence and correctness of these facts. This evaluation design rewards dataset-backed outputs, rather than factual recall, and thus aligns with the usage of AI co-scientists.

We instantiate the HEUREKABENCH framework in the domain of single-cell biology, resulting in sc-HEUREKABENCH, a benchmark with 50 OEQs and 50 MCQs across 41 validated insights. Using our benchmark and evaluation, we compare a suite of current state-of-the-art single-cell biology agents in answering open-ended questions on well-established findings from experimental single-cell RNA-seq (scRNA-seq) datasets. Furthermore, we systematically analyze three core components of an agent − *planner*, *critic*, and *retriever* − and quantify their impact in the context of our benchmark. Notably, we find that closed-source models, as planners, exhibit better performance than open-

source counterparts; however, with multi-turn reasoning, the gap can be narrowed. Additionally, incorporating a *critic* at the end of the agent loop detects and revises ill-formed responses and analyses. Together, our benchmarking framework HEUREKABENCH and the sc-HEUREKABENCH benchmark, along with insights into current agent designs, offer tools to advance the development of co-scientists in single-cell biology and provide the path for designing benchmarks for solving open-ended scientific problems in other scientific domains.

## 2 RELATED WORK

**Agents for scientific discovery.** The increasing accessibility of LLMs and open-source frameworks to integrate them with external environments and custom tools (Chase, 2022; Narayanan et al., 2025) has spawned multiple agents to either assist in or automate scientific discovery. Some examples include Robin (Ghareeb et al., 2025), which can repurpose existing drugs as potential candidate therapeutics for other diseases, and SciAgents (Ghafarollahi & Buehler, 2025), which can traverse knowledge graphs to propose hypotheses about previously unrecognized relationships between scientific concepts. In computational chemistry, ChemCrow (M. Bran et al., 2024) and LLM-RDF (Ruan et al., 2024) can synthesize new compounds and guide end-to-end chemical synthesis. Cactus (McNaughton et al., 2024) can tackle drug and molecular property prediction tasks with domain-specific tools. Further, AstroAgents (Saeedi et al., 2025) can process mass spectrometry data to generate plausible hypotheses in the context of existing astrobiology literature.

Besides these domains, a number of agentic workflows have been recently proposed for single-cell biology at different levels of specificity. BioDiscoveryAgent (Roohani et al., 2025) is designed to propose gene perturbation panels to achieve a target cell phenotype. CellAgent (Xiao et al., 2024) involves a planner and executor to generate code blocks, and an evaluator to evaluate the output for three well-defined tasks. However, an agent as a co-scientist should build end-to-end workflows supporting the generation of novel open-ended hypotheses. To this end, CellVoyager (Alber et al., 2025), BixBench-Agent (Mitchener et al., 2025), and Biomni-A1 (Huang et al., 2025b) plan autonomous workflows to derive data-driven insights. The key differences include the environment construction and agentic architecture. However, such agents are currently evaluated on more straightforward scientific reasoning and computational questions, rather than their intended use case.

**Benchmarking agents for data-driven scientific discovery.** An increase in agents proposed across domains necessitates suitable benchmarks to evaluate the progress. Existing benchmarks for scientific agents share some common characteristics. First, several benchmarks assess scientific thinking in a standalone manner. CORE-Bench (Siegel et al., 2024) creates tasks to test reproducibility in code repositories. LAB-Bench (Laurent et al., 2024) and HLE (Biomedicine) (Phan et al., 2025) tests general biological knowledge and reasoning. SciCode (Tian et al., 2024) evaluates code generation from scientific concepts. Yet none of them pose tasks suitable for evaluating co-scientists. Second, data-driven single-cell benchmarks (Majumder et al., 2025; Mitchener et al., 2025; Chen et al., 2025) primarily involve statistical or computational problems, and others, such as BaisBench (Luo et al., 2025), are limited to multiple-choice questions alone. As a result, evaluation is constrained to matching against fixed answer sets, which does not reflect the open-ended, exploratory nature of scientific discovery. In addition, BaisBench generates research questions automatically using a single LLM without human oversight, reducing its reliability. Gu et al. (2024a) also creates multiple-choice questions only, but improves reliability by crowdsourcing annotations of intermediate analysis decisions, and evaluates the open-endedness of scientific discovery by matching these decisions. As a result, the evaluation is constrained to intermediate human preferences and ignores the final agent interpretation. In contrast, our HEUREKABENCH framework utilizes a multi-LLM pipeline where question generation is divided into suitable subtasks with human supervision. Furthermore, we create both open-ended and multiple-choice research questions which require agents to plan workflows that involve multiple computational steps and conclude with an open-ended, data-driven interpretation, which is evaluated against ground-truth answers from scientific findings.

## 3 HEUREKABENCH FRAMEWORK

We introduce HEUREKABENCH, a benchmarking framework designed to evaluate the potential of LLM-based agents as co-scientists. Whereas prior benchmarks target isolated subtasks such as fac-

tual recall, tool use, or single-step computation, HEUREKABENCH unifies them within a single framework. Its research-oriented questions require agents to analyze datasets and derive insights that are not explicitly present in the input and cannot be retrieved from general knowledge alone. Solving these questions is a multi-step reasoning process that includes selecting appropriate analyses, interpreting results, and reasoning over evidence, thereby closely reflecting the open-ended problem-solving process of scientific discovery.

In the remainder of this section, we *(i)* formalize the co-scientist task, *(ii)* describe the HEUREKABENCH framework, *(iii)* propose evaluation strategies, and *(iv)* highlight domain-specific decisions and challenges to instantiate the framework in single-cell biology and create sc-HEUREKABENCH.

## 3.1 CO-SCIENTIST TASK OVERVIEW

An agent as a co-scientist is expected to handle exploratory research questions grounded in real experimental data. To evaluate such systems, we formalize the benchmark task as a collection of triplets $(D, Q, A)$, where $D$ is a dataset, $Q$ a research question, and $A$ the corresponding ground truth answer. In this setup, each **dataset** ($D$) consists of experimental datasets from a specific scientific domain and can include auxiliary files. For example, in single-cell biology, $D$ might consist of a gene count matrix derived from the wet-lab experiment, alongside metadata describing the treatment conditions. Next, each associated **question** ($Q$) corresponds to an open-ended research question that demands multi-step reasoning over $D$. An example from single-cell biology would be a question that asks *what changes in cytokine expression are observed in the aging muscle microenvironment*, requiring both analysis and interpretation. Finally, a ground-truth **answer** ($A$) is needed to appropriately judge agent responses. This triplet formulation enforces the essential components of real-world co-scientist tasks: *(i)* authentic experimental datasets, *(ii)* open-ended research questions requiring workflow-level reasoning, and *(iii)* scientifically validated answers for evaluation. Together, they ensure that performance on the benchmark reflects the agent's ability to function as a scientific collaborator, rather than an isolated problem solver.

## 3.2 HEUREKABENCH CREATION

To build $(D, Q, A)$ triplets, we leverage published research studies. We develop an LLM-based, multi-step pipeline that processes scientific publications alongside their associated datasets and code repositories. The foundation of the HEUREKABENCH are the *high-level scientific insights*, which correspond to novel findings in the studies, and are used to establish research-oriented questions. The creation pipeline has two main stages: *(i)* insights generation, presented in Figure 1(a), where many candidate insights are generated and validated semi-automatically, and *(ii)* questions generation, presented in Figure 1(b), where validated insights are formulated as $(Q, A)$ pairs. In the following, we detail how each stage operates and how these together translate published research studies into co-scientist benchmark data instances.

**Insights generation.** The first stage of the HEUREKABENCH pipeline extracts and validates scientific insights from published studies using their code as a means of validation to retain only reproducible insights. To achieve this, we design a modular pipeline: *InsightExtractor* proposes candidate insights from the paper, while *CodeDescriber* converts code scripts into natural language summaries. The outputs of these modules are combined via *CodeMatcher*, which links insights to the most relevant code descriptions and retrieves scripts that could support the insight. Finally, *CodeGenerator* composes these scripts into a multi-step workflow for each candidate insight.

Using *InsightExtractor* module, we represent each insight by three linked components: *(i)* a *summary* providing an accessible description, *(ii) experimental techniques* mentioned in the paper that are used to establish the insight, and *(iii) grounding text* that uses verbatim statements from the paper relevant to the insight as supporting evidence. This structured representation organizes insights and their evidence, guiding all subsequent steps in HEUREKABENCH framework. Prompts for each module are provided in Appendix A.1.

After generating candidate insight-code workflow pairs, human reviewers run the code. The reviewers can apply minor code adjustments (*e.g.*, renaming columns to align with the dataset) or propose supplementary files necessary to fully validate the insight. Once the code runs successfully, they

verify the reproducibility of each insight by ensuring that the obtained results, such as figures or statistics, match those reported in the insight's *grounding text* and the study.

The output of this stage is a pool of validated insights. Thus, our framework filters unverifiable insights, directly addressing the reliability gap of co-scientist benchmarks that rely solely on single LLM capabilities (Luo et al., 2025).

**Questions generation.** For each validated insight, we generate two question types: *(i) open-ended questions (OEQs)* and *(ii) multiple-choice questions (MCQs)*. OEQs represent the primary format, reflecting real-world research, which rarely offers fixed alternatives and instead requires synthesizing evidence, constructing reasoning chains, and articulating conclusions in free-form language. They assess whether an agent can perform genuine data-driven reasoning and insight discovery rather than rely on recognition or elimination strategies. In contrast, MCQs provide a more constrained yet informative evaluation setting for rapid agent prototyping. Note that the MCQ group includes questions with both single and multiple correct options.

The generation process for both QA types follows a similar pipeline, differing only in the prompting strategies for automatic generation and the subsequent filtering steps. We employ few-shot prompting to generate two $(Q, A)$ pairs for each insight (including its *summary*, *experimental techniques*, and *grounding text*). For MCQs, we emphasize creating challenging distractors that capture plausible misinterpretations or common analytical errors, ensuring that correct answers require a genuine understanding of the data. OEQs, on the other hand, are intentionally less specific, allowing multiple approaches to reach the correct answer. The prompts are available in Appendix A.2. Following the generation, questions undergo a two-stage filtering process: *(i)* automatic filtering to remove easy questions solvable using LLMs' pretraining knowledge and *(ii)* manual review to remove hallucinations, duplicates, and questions based on non-validated parts of the insights. Additional details about filtering can be found in Appendix A.4.

## 3.3 EVALUATION

Evaluating agents on our benchmark poses distinct challenges for both question types. For open-ended research questions, agent analyses may uncover additional conclusions beyond the annotated ground truth. Moreover, in natural-language responses, an agent could potentially rely on prior knowledge from the literature rather than exploring the dataset. Therefore, evaluation of OEQs must go beyond surface-level matching. We adopt G-Eval (Liu et al., 2023) with GPT-4o as the LLM-Judge, assigning ratings between 1 and 5. To ensure scientific rigour, we instruct the judge to first decompose both the response and the ground truth into atomic facts (*e.g.*, conditions, trends, conclusions) and then assess overlap across complete, partial, and missing facts (the entire rubric is provided in Appendix B). An agent receives the highest score only if all ground-truth facts are present and no contradictions occur, while additional non-conflicting findings are not penalized. An illustration of agent evaluation on open-ended questions is presented in Figure 1(c).

For MCQs, we report accuracy as the primary metric. Although all choices are generated solely from the structural representation of validated insights (including their *summary*, *experimental details*, and *grounding text*), some options marked as incorrect may still appear scientifically plausible due to being LLM-generated. To account for such scenarios, we also report precision and recall.

## 3.4 INSTANTIATING HEUREKABENCH IN SINGLE-CELL BIOLOGY

We instantiate the HEUREKABENCH framework in single-cell biology, particularly for studies on the analysis of scRNA-sequencing datasets (Tang et al., 2009). First, we curate a pool of 22 papers published in *Nature* and *Cell* journals in 2024 and 2025, with corresponding open-source code repositories and open-access datasets from the CellxGene (CZI Cell Science Program et al., 2025) or publication resources. Our choice to restrict to recent publications partially mitigates the risk that agents can rely solely on memorized knowledge. We then applied our insight generation pipeline, utilizing GPT-4o (Achiam et al., 2023) in *InsightExtractor* and Claude-4-Sonnet (Anthropic, 2025a) in the remaining code modules, to produce 10 candidate insights per paper and retained only those that could be validated. This process yielded a final pool of 41 validated insights across 13 papers (nine from *Nature* and four from *Cell*; listed in Appendix C.1). We treat an insight as validated only if the workflow output reproduces the results reported in the paper. While other generated insights may be

validated with additional information, we treat them as invalidated within our framework. We provide additional discussion about tasks, publications, and invalidated insights in Appendix C.2, C.3, and C.4 respectively. Finally, using our question generation pipeline, we derived 50 OEQs and 50 MCQs from the validated insights, constructing the **sc-HEUREKABENCH**, the single-cell biology instantiation of our framework.

### 3.4.1 SC-HEUREKABENCH-TOOLUSAGE BENCHMARK

During manual workflow review, we observed multiple insights that relied on domain-specific tools and databases (*e.g.*, SCENIC (Aibar et al., 2017), CellPhoneDB (Efremova et al., 2020), CellChat (Jin et al., 2021)) as well as machine learning methods (*e.g.*, Non-negative Matrix Factorization). These insights could not be validated because the *CodeGenerator* hallucinated the usage of these tools or their outputs. Nevertheless, they can serve as a specialized benchmark for evaluating agents' ability to leverage domain-specific tools, particularly when selected from papers containing other validated insights. We created 12 OEQs from our collection of 13 papers that included at least one validated insight, excluding questions that relied on imaging or spatial transcriptomics tools. These questions constitute the sc-HEUREKABENCH-ToolUsage (sc-HEUREKABENCH-TU) benchmark, suitable for evaluating agents with access to domain-specific tools (Huang et al., 2025b).

## 4 EXPERIMENTS

Within our experiments, we first collect proxy datasets to evaluate the insights construction pipeline of our proposed HEUREKABENCH framework. Second, we compare existing agents in single-cell biology to act as co-scientists on the sc-HEUREKABENCH benchmark. Next, we discuss the impact of various design choices within the agent on its capabilities as a co-scientist. We also perform an alignment study between the scores assigned by GPT-4o and other closed-source LLM-based judges, along with human graders, to increase the reliability of our evaluation framework.

### 4.1 EVALUATION OF INSIGHT CONSTRUCTION

The insight construction stage within the HEUREKABENCH framework comprises four modules, as described in Section 3.2. To evaluate the *InsightExtractor* module, we assemble pairs of open-access publications linked with expert findings from two resources. Specifically, we leverage FlyBase (Öztürk-Çolak et al., 2024), an openly accessible genome database of *Drosophila*. We focus on 10 random gene identifiers to scrape 50 pairs of publications and corresponding expert findings. The list of genes and other collection details is provided in Appendix D.1. In addition to FlyBase, we also repurpose BixBench (Mitchener et al., 2025) to obtain a list of 21 publications and expert hypotheses pairs. We run our InsightExtractor module on each paper and use GPT-4o as a judge to label if our generated insights are *strongly related*, *weakly related*, or *unrelated* to the expert insight. The instructions for the

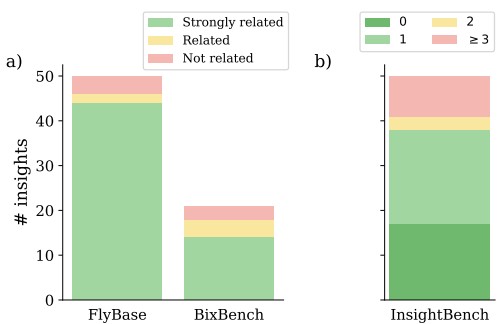

Figure 2: (a) Evaluation of the *InsightExtractor* module. Number of insights related to expert findings in FlyBase and BixBench. (b) Evaluation of the *CodeDescriber* and *CodeMatcher* modules. Number of insights per number of incorrectly retrieved files. 0 indicates all files retrieved correctly. Red indicates failure cases.

judge are available in Appendix D.2. We show our results in Figure 2(a). For FlyBase, we obtain 44 *strongly*, two *weakly*, and four *unrelated* findings. At the same time, for BixBench, we have 14 *strongly*, four *weakly*, and three *unrelated*. Hence, *InsightExtractor* module is capable of determining expert-level insights from scientific publications.

Furthermore, we collect pairs of expert-created insights and corresponding multi-step workflows from InsightBench (Sahu et al., 2025) to assess our *CodeDescriber* and *CodeMatcher* modules. We convert each step within a workflow into a code file and create a code repository with these files. We restrict the repository to 50 insights (and 215 code files) to match the maximum size of a real-world

repository from our set of validated publications in the sc-HEUREKABENCH[1]. The *CodeDescriber* annotates the code scripts, and given the insight description, *CodeMatcher* retrieves the relevant files. Although this is a challenging setting, out of 215 scripts, 158 scripts were correctly matched to the respective insights, and an average of **74.6**% of files are retrieved correctly across all insights. Additionally, we also plot the number of insights per incorrectly retrieved files in Figure 2(b), which shows that our modules can select the relevant files corresponding to insights for code generation.

## 4.2 EVALUATION OF SINGLE-CELL BIOLOGY AGENTS

We benchmark three state-of-the-art agents for single-cell biology on the sc-HEUREKABENCH: Biomni (Huang et al., 2025b), CellVoyager (Alber et al., 2025), and BixBench-Agent (Mitchener et al., 2025). Biomni (Huang et al., 2025b) comprises Biomni-E1, an extensive biomedical environment that contains 105 software packages, 59 databases, and 150 specialized biomedical tools, including cell biology and genomics domains, and Biomni-A1, an agent that navigates Biomni-E1 to build autonomous multi-step workflows for data analysis on experimental data. CellVoyager (Alber et al., 2025) is designed to propose novel hypotheses from existing studies and validate them. However, unlike Biomni, it follows a rigid architecture, where each step contains specific LLMs for sequentially planning, coding, handling feedback, interpreting outputs, and summarizing. Consequently, we remove the hypothesis proposal step and instruct to answer research questions. Finally, BixBench-Agent (Mitchener et al., 2025) is a more black-box agent that relies on the Aviary custom environment (Narayanan et al., 2025), with predefined bioinformatics packages.

We observed a few issues through initial runs of BixBench-Agent and CellVoyager. Specifically, CellVoyager, due to its design, exhibited significant time and API cost requirements, taking up to an hour to answer specific questions, while BixBench-Agent crashed on large datasets. To mitigate computational costs and have fair comparisons, we report results on questions related to datasets smaller than 750 MB, which we term sc-HEUREKABENCH-Lite. This subset contains 22 out of 50 OEQs and 18 out of 50 MCQs on which all agents could run. The similarity of tasks between sc-HEUREKABENCH-Lite and sc-HEUREKABENCH is discussed in Appendix D.3 and Figure 5.

Table 1: Evaluation of single-cell agents on sc-HEUREKABENCH-Lite. All agents use Claude-4-Sonnet LLM. Best and second best performance is **bolded** and underlined respectively. Higher values are better for all metrics. The task prompt for agents is available in Appendix G.

| | OEQs | MCQs | | |
|---|---|---|---|---|
| **Agent** | Correctness [1-5] | Accuracy [%] | Recall [%] | Precision [%] |
| BixBench-Agent | **2.34** | 44.44 | 80.56 | 62.96 |
| CellVoyager | 2.03 | 27.78 | 38.89 | 32.41 |
| Biomni | 2.31 | **50.00** | **88.24** | **76.96** |

Results in Table 1 reveal that BixBench-Agent and Biomni outperform CellVoyager in both formats, which indicates that a more flexible agent loop is capable of building robust workflows and answering research questions. Biomni also contains more domain-specific tools and databases compared to other agents. Meanwhile, a close inspection of CellVoyager outputs reveals restrictive code-fixing capabilities and difficulty in incorporating multiple feedback in each step as the main reasons for reduced performance. Further, the number of steps needs to be pre-specified (we use eight instead of the default six), which sometimes limits the workflow to finish and output an appropriate answer.

## 4.3 IMPACT OF DESIGN CHOICES ON CO-SCIENTIST

Within our definition of agent as a co-scientist, three key components, including *planner*, *critic*, and *retriever*, provide specific capabilities that can influence its performance. To assess their individual contributions, we introduce and discuss their roles below and perform targeted ablation studies. For this analysis, we focus on Biomni as the only competitive model that could effectively and efficiently run across all datasets, as shown in Table 1.

---

[1]Out of the initial 22 papers, 18 have less than 100 code files

### 4.3.1 PLANNER ABLATIONS

The planner is responsible for generating the initial plan when an agent is provided with experimental data and a research question. Subsequently, as the agent loop continues, its role expands to receive feedback from the external environment (*e.g.*, results, error messages, etc.) and potentially modify the plan. Generally, strong reasoning models are selected as a planner. In Biomni, the planner can decide to generate either a plan or code actions as required. In our experiments, we compare a range of open- and closed-source LLMs, further differentiating open models by scale and reasoning style (*thinking* vs. *non-thinking*). Results on sc-HEUREKABENCH are shown in Table 2.

Amongst all LLMs, Claude-4-Sonnet (Anthropic, 2025a) achieves the highest overall performance across OEQs and MCQs. Particularly in OEQs, it outperforms the second-best model with a significant margin (2.58 against 2.08), highlighting the benefits of frontier closed-source models. Within the Qwen (Yang et al., 2025) model family, performance consistently improves with increasing model size, which suggests that model parameter scale is crucial, and the *thinking* variant provides additional gains in correctness on OEQs due to its better reasoning abilities (+0.28 from non-thinking and +0.38 from smaller model). Among open-source models, the best-performing model was GPT-OSS-120B (Agarwal et al., 2025), a recent OpenAI model designed for use within agentic workflows. Overall, these ablations confirm that agent performance as a co-scientist is highly dependent on model family, scale, and reasoning style.

Table 2: Planner ablation results on sc-HEUREKABENCH with Biomni agent. Best and second best performance is **bolded** and underlined respectively. Open denotes if the LLM is open- or closed-source. Correctness scores are averaged across three independent agent runs. Accuracy, recall, and precision metrics are denoted in %. Higher metric values are better.

| Model | Open | OEQs | MCQs | | |
| | | Correctness [1-5] | Accuracy | Recall | Precision |
|---|---|---|---|---|---|
| MedGemma-27B | ✓ | $1.53 \pm 0.02$ | 20.41 | 41.84 | 37.59 |
| Qwen3-32B | ✓ | $1.47 \pm 0.02$ | 40.00 | 59.50 | 55.50 |
| Qwen3-235B | ✓ | $1.57 \pm 0.06$ | 42.00 | 64.50 | 61.00 |
| Qwen3-235B-THINKING | ✓ | $1.85 \pm 0.03$ | **46.00** | 65.00 | 57.33 |
| GPT-OSS-120B | ✓ | $2.08 \pm 0.05$ | 42.00 | 52.00 | 47.00 |
| GPT-4o | ✗ | $1.68 \pm 0.05$ | 18.00 | 59.00 | 44.83 |
| Claude-4-Sonnet | ✗ | **$2.58 \pm 0.05$** | 44.00 | **85.00** | **66.33** |

**Inter-rater alignment study between LLM-based judges.** We perform an alignment study between the correctness scores assigned by three different closed-source LLMs (including the proposed GPT-4o) as judges. In particular, we select Claude-4.5-Sonnet (Anthropic, 2025b) and Gemini-2.5-Pro (Comanici et al., 2025), which are not used as planner LLMs within the Biomni agent and belong to a different model series than GPT. We consider the three best-performing planner models, *i.e.*, Claude-4-Sonnet, GPT-OSS-120B, and Qwen3-235B-THINKING and show the ranks assigned by the three judges in Figure 3. We observe that all judges agree on the ranking of the planner models. Next, for each model, we compute more fine-grained inter-rater agreement metrics comparing the correctness scores assigned by GPT-4o and other judges. In particular, we report Spearman's rank correlation and unweighted Cohen's kappa ($\kappa$) with quadratic

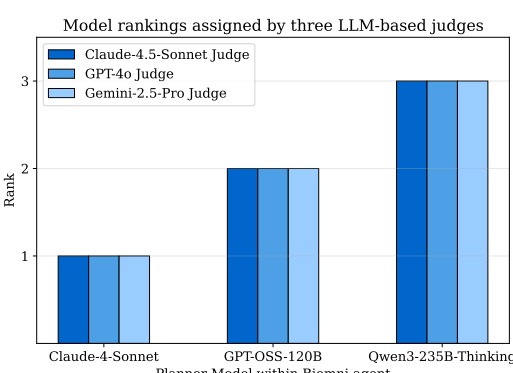

Figure 3: Rankings of the three best-performing planner LLMs within the Biomni agent as per three closed-source LLM-based judges. The plot shows that the three judges agree on the performance of planner models and select Claude-4-Sonnet as the best planner model.

penalty, which are suitable for ordinal discrete ratings. We find that the mean correlation across three planner models between GPT-4o and Claude-4.5-Sonnet judges is $0.84_{\pm 0.03}$, whereas it is $0.79_{\pm 0.01}$ with Gemini-2.5-Pro. Similarly, the $\kappa$ is $0.81_{\pm 0.03}$ and $0.71_{\pm 0.04}$ respectively. This study indicates high agreement between different LLM judges. We provide detailed values of these metrics in Appendix D.4 and use GPT-4o for all subsequent evaluations.

**Human-raters alignment study with GPT-4o.** We additionally carry out an alignment study comparing the LLM-judge scores with expert human assessments. Consequently, we selected 25 open-ended answers from Biomni (GPT-OSS-120B) and asked 11 human experts to assign a score between 1 and 5 (inclusive) for each agent response relative to the ground truth. We define our human experts as Ph.D. students or post-doctoral researchers with at least 1 year of experience working with single-cell data, including analyzing and drawing conclusions from it. The experts are from four different universities and six different labs. We aggregated expert scores using both mode and median for each question. The difference between human and LLM-judge correctness scores was $\leq 1$ for $92\%(23/25)$ and $96\%(24/25)$ of questions for mode and median, respectively, indicating strong agreement. Furthermore, Spearman's rank correlation between expert and LLM-judge scores was $0.93$ (mode) and $0.90$ (median), showing a strong association between both ratings. Finally, we also computed $\kappa$ and obtained $0.85$ for both aggregation methods.

### 4.3.2 CRITIC ABLATIONS

While closed models currently dominate, we wondered whether the performance of open-source LLMs can be improved by using a critic. Specifically, the critic provides critical recommendations on the outputs at different stages of the agent loop. Typically, an LLM is used as a critic, which takes the relevant outputs and generates actionable feedback. In our experiments, we compare different positions for the critic module. **No-critic** denotes the absence of a critic LLM and is the default mode of Biomni. Additionally, we compare **Plan-critic** or **End-critic**, where the critic is present immediately after the planner drafts an initial plan or at the end of analysis when the planner decides to exit. Thus, the feedback can either request to modify the initial plan or suggest a new workflow to address key missing parts in the prior analysis. This draws parallels to the real world, where a scientist can get a second opinion from a colleague on an initial idea or on the results of an analysis.

Table 3: Critic ablation results on sc-HEUREKABENCH (OEQs) with Biomni agent. Reported are counts of categories judged as better or worse, as well as average scores before and after adding the critics. Also noted the number of questions per score category. Correctness scores for entire sc-HEUREKABENCH benchmark are averaged across three independent agent runs (denoted with All). Other analyses are for a single agent run. #q stands for the number of questions.

| Score cat. | # q | No-critic Correctness [1-5] | End-critic Correctness [1-5] | End-critic Better / Worse | Plan-critic Correctness [1-5] | Plan-critic Better / Worse |
|---|---|---|---|---|---|---|
| | | | **GPT-OSS-120B** | | | |
| High | 4 | 4.55 | 4.85 | 0 / 0 | 2.00 | 0 / 3 |
| Mid | 16 | 2.79 | 2.98 | 6 / 5 | 2.44 | 4 / 9 |
| Low | 30 | 1.32 | 1.91 | 10 / 4 | 1.62 | 7 / 5 |
| All | 50 | 2.04 (2.08 ± 0.05) | 2.49 (2.40 ± 0.08) | 16 / 9 | 1.91 (1.92 ± 0.10) | 11 / 17 |
| | | | **Qwen3-235B-THINKING** | | | |
| High | 4 | 4.67 | 3.64 | 0 / 2 | 3.15 | 0 / 3 |
| Mid | 13 | 2.66 | 2.09 | 1 / 8 | 1.92 | 3 / 8 |
| Low | 33 | 1.20 | 1.48 | 6 / 2 | 1.35 | 6 / 2 |
| All | 50 | 1.86 (1.85 ± 0.03) | 1.81 (1.73 ± 0.09) | 7 / 12 | 1.65 (1.56 ± 0.08) | 9 / 13 |

To better understand the influence of the critics, we categorize the original scores into high-, mid-, and low-performing questions. Specifically, scores above four are labeled as *high*, scores above two as *mid*, while other scores are labeled as *low*. This categorization enables us to examine which groups of initial scores were affected by the ablation experiment and in what manner. Alongside reporting category-wise correctness, we also count the number of questions that achieved better or worse scores following the ablation, where the difference exceeded 0.5 in either direction.

Table 3 reports critic ablations with the two best-performing open-source LLMs. We report correctness scores on entire sc-HEUREKABENCH (termed as All) averaged over three independent runs, and categorize answers from one run per setup for deeper analysis. For GPT-OSS-120B, the End-critic yields the consistent gains in performance, raising average correctness scores up to $2.49$ (close to $2.58$ with Claude-4-Sonnet in Table 2), with the strongest effect on low-scoring questions ($+0.6$ over 30 cases). For Qwen3-235B-THINKING, End-critic helps on low-quality answers but degrades stronger ones, meaningfully improving only seven questions in total compared to 16 for GPT-OSS-120B. In contrast, the Plan-critic consistently reduces mid- and high-scored questions, leading to overall drops of $0.13$ and $0.19$ points for GPT-OSS and Qwen3, respectively. These results indicate that critic placement is crucial – feedback at the end can improve poorer responses, while in the beginning can potentially disrupt the reasoning trajectories. However, the extent of benefit and disadvantage depends on the underlying LLM. Furthermore, the content and quality of feedback are influenced by the critic LLM and, more importantly, the responses generated by the planner, which adds stochasticity to the final answers and workflows generated by agentic systems with a critic.

### 4.3.3 RETRIEVER ABLATIONS WITH SC-HEUREKABENCH-TU

The retriever module within Biomni is an LLM tasked with selecting the appropriate set of tools, software, and databases relevant to solving the task before the initial plan is created. This intermediate step avoids overwhelming the planner with the extensive set of tools available in Biomni-E1. To properly evaluate the importance of a retriever, we focus on weaker open model agents and sc-HEUREKABENCH-TU, where the agent requires access to domain-specific tools to provide a well-formed answer. In Table 4, we provide the results with the retriever module disabled, averaged across three independent agent runs. Although the correctness scores could vary due to the small size of the sc-HEUREKABENCH-TU (12 OEQs), we observe a significant drop in correctness for both models across multiple agent runs, indicating that without the retriever, the agent is unable to choose the proper set of tools and submits a suboptimal response.

Table 4: Retriever ablation results on sc-HEUREKABENCH-TU. Reported performance of the Biomni agent with and without the retriever. Correctness scores are averaged across three independent agent runs. A higher correctness score indicates better performance.

| | Correctness [1-5] | |
|---|---|---|
| **Agent** | With retriever | No retriever |
| GPT-OSS-120B | $2.15 \pm 0.09$ | $1.56 \pm 0.22$ |
| Qwen3-235B-THINKING | $1.92 \pm 0.13$ | $1.80 \pm 0.07$ |

## 5 CONCLUSION AND FUTURE WORK

In this work, we introduce HEUREKABENCH, a framework for constructing benchmarks to evaluate agents as AI co-scientists. The framework leverages scientific publications and their associated codebases to generate open-ended research questions that require exploration of experimental datasets, multi-step workflows, and reasoning to produce data-backed responses. We also propose an evaluation paradigm using an LLM-as-a-judge, designed to appropriately score such free-form scientific outputs. We instantiate the framework in single-cell biology as sc-HEUREKABENCH, comparing the performance of published biological agents on the benchmark and exploring several potential training-free improvements. These include incorporating a critic module, which in our experiments allowed an open-source model to achieve performance comparable to that of a closed-source model, and using LLMs optimized for agentic tasks, which consistently yielded better results.

For future work, we propose adapting the framework to other scientific domains, which would require domain experts during manual validation. Additionally, LLM-based agents could be employed to validate intermediate outputs against published studies, thereby adding rigour to the pipeline. Furthermore, the sc-HEUREKABENCH can be continuously evolved with new scientific publications to update questions that can be used to evaluate new pre-trained LLM-based agents. A current limitation of our work is that evaluation relies solely on the final agent response. This could be improved by implementing a strategy that verifies intermediate workflow steps and assigns partial credit for correct steps. Finally, performance could be further enhanced by leveraging open-source LLMs trained for agentic tasks and incorporating suitable architectural modifications, suggesting that future efforts focus on post-training LLMs for their intended role as AI co-scientists.

ACKNOWLEDGMENTS

We thank Artyom Gadetsky, Maxim Kodryan, Tingyang Yu, Yulun Jiang, Debajyoti Dasgupta, Haris Kupinić, Vijay Baskar, Lloyd Steele, and Muzlifah Haniffa for their valuable suggestions and discussions, which helped improve the clarity of the manuscript. Further, we are immensely grateful to Tianze Wang, Shuo Wen, Luca Fusar Bassini, Alireza Gargoorimotlagh, Wei Qiu, Maurizio Fiusco, Tingyang Yu, Lucas Miranda, Xiaojian Chen, Maciek Wiatrak, David Frühbuß, and Camille Lambert for judging agent responses to our benchmark questions within the context of human evaluations. Computational resources used in the work were provided by EPFL. We gratefully acknowledge the support of the Swiss National Science Foundation (SNSF) starting grant TMSGI2_226252/1, SNSF grant IC00I0_231922, and the Swiss AI Initiative. M.B. is a CIFAR Fellow in the Multiscale Human Program.

ETHICS STATEMENT

Our work involves evaluating LLM-based agents and their potential as AI co-scientists. Our benchmark is created using open-access, peer-reviewed scientific publications and open-source code repositories on GitHub, and it contains research questions that prompt the agents to explore the public experimental datasets associated with these. However, since LLM-based agents are autonomous systems that are allowed to generate code snippets, they can potentially generate malicious code snippets that affect the user's system. When using our benchmark, we recommend that users run the agent evaluation within suitable sandbox environments that incorporate necessary safety measures.

REPRODUCIBILITY STATEMENT

To support reproducibility of our work, we provide the agent inference and evaluation codebase alongside our benchmarks, with detailed instructions for using our proposed benchmark creation framework, HEUREKABENCH, at `https://github.com/mlbio-epfl/heurekabench`.

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

APPENDIX

## A    DETAILS ON HEUREKABENCH FRAMEWORK

### A.1    PROMPTS FOR INSIGHT GENERATION

The first stage of our proposed HEUREKABENCH framework is insights generation, which consists of four modules − *InsightExtractor*, *CodeDescriber*, *CodeMatcher*, and *CodeGenerator*. Each of these modules is based on an LLM as described in Section 3.2 and Section 3.4. Below, we provide their prompts.

---

**InsightExtractor prompt**

**You are a scientific research assistant with expertise in interpreting and analyzing high-impact scientific publications.**

You will be provided with a research article in the field of single-cell RNA sequencing biology that presents novel findings.

Your task is to extract **10 distinct, non-overlapping key insights** grounded specifically in the **authors' analysis and interpretation of their data**, rather than general background or established biological facts. Each insight must be **analytically derived**—emphasizing conclusions, patterns, or implications the authors draw from their results, demonstrating a deep understanding of the study's significance.

**What is a "High-Level Insight"?**

A high-level insight is a concise, meaningful takeaway capturing a core contribution or finding of the study. Such insights typically appear in:

- The abstract
- Discussion or conclusion sections
- Summaries within results or figure legends
- Syntheses of experimental findings

These insights should avoid vague or broad restatements. Instead, they should clarify **what** was found, **why** it matters, and **how** the authors arrived at the conclusion.

**Task Instructions:**

Extract and rank **10 insights** by importance, using this structured format for each:

**Insight #X**
*Summary:* A clear, concise (1–3 sentences) paraphrased summary of the insight, capturing a key finding, interpretation, or contribution.
*How it was derived:* A brief paragraph (3–5 sentences) detailing how the insight was obtained, focusing on information sufficient to reproduce the analysis. Include:

- Experimental and computational methods used
- Key data trends, statistical analyses, or comparisons
- Supporting figures, tables, or quantitative evidence, if applicable
- Authors' interpretations relevant to the insight
- Reference the relevant paper sections (*e.g.*, Results, Figures, Abstract)

*Relevant text paragraphs:* Up to 10–15 sentences from the paper that underpin the insight, for context. Replicate the original text as closely as possible, ensuring it is clear and informative. This should reflect the authors' own words and interpretations, not your paraphrasing.

---

**Content Prioritization**

When reading the article, prioritize these sections in order:

1. Abstract — overarching goals and headline findings
2. Main Results — detailed data, trends, and discoveries
3. Figures and Figure Legends — visual summaries and experimental design
4. Discussion — interpretations, implications, future directions
5. Methods — how insights were generated

**Additional Guidelines**

- Use paraphrasing to avoid direct quotes in summaries and derivations.
- Ensure each insight stands alone and is understandable without the full paper.
- Favor insights that:
    - Reveal cause-effect relationships
    - Highlight unexpected or counterintuitive results
    - Synthesize multiple lines of evidence
    - Introduce novel techniques or conceptual advances
- Exclude formatting artifacts (page numbers, citation codes, etc.).
- If the study has multiple sub-experiments or datasets, derive at least one insight from each.
- Do **not** fabricate or simulate insights not explicitly present in the paper.
- Your audience is a biomedical researcher, so maintain rigor and accuracy.

**Example Output (illustrative only):** *[an example]*

Now, carefully review the article and **generate 10 insights** using this structure and guidelines.

---

**CodeDescriber prompt**

You are a senior research-software analyst.
Task: You will receive N source-code files, each delimited like this:

```
### BEGIN <relative/path/to/file.ext>
<full file content>
### END <relative/path/to/file.ext>
```

For **each file** produce a single, well-structured paragraph (3-6 sentences) that:

- names the main functions / classes / entry points
- states the scientific or analytic goal the script helps achieve
- notes crucial implementation details (*e.g.* I/O formats, key algorithms, dependencies, or domain-specific nuances)

Output format:
Return one JSON dictionary whose keys are the *exact* file paths and whose values are your paragraphs, *e.g.*

```
{
  "analysis/load_data.R": "This script ...",
  "simulation/core.py":   "This module ..."
}
```

**CodeMatcher prompt**

You are an expert research assistant helping to link biological research insights with relevant analysis scripts.
You are given:

1. A **High-Level Insight**, which includes:
    - A **summary**, capturing the main biological finding or claim.
    - A **description**, detailing how the insight was derived — including techniques (*e.g.* scRNA-seq, UMAP, clustering), key genes or cell types involved, and types of visualizations or computational analyses mentioned.
    - A **relevant text** section, which may include parts from the paper that provide context or support for the insight. Use the associated paragraphs to identify any additional details that could help you in retrieving relevant code and creating code snippets.

2. A list of **code files**. Each file has:
    - A **file path**
    - A **description** of what the script does, including major operations (*e.g.* PCA, UMAP, heatmaps), the cell types or conditions it analyzes, and its purpose (*e.g.* visualization, clustering, gene expression comparison).

Definition of "High-Level Insight": A high-level insight is a concise but meaningful takeaway that captures one of the central contributions or findings of the study. These are the types of statements you might expect to find in:

- The abstract
- The discussion or conclusion
- Summaries in results or figure legends
- A high-level synthesis of experimental findings - Such insights would not be vague restatements of a section but would reflect the what, why, and how of a meaningful result or observation.

Task instructions:

- Carefully read the insight's description and match it with the most relevant code files based on the type of analysis, data focus (*e.g.* B cells, T cells), and outputs mentioned.
- Return only a valid Python list of up to 5 string file paths that are most relevant to the insight above. Do not include explanations, just the list.
- Also, avoid faking or simulating file paths. Your user is a biomedical researcher and expert programmer. Thus, stay true and rigorous.

**CodeGenerator prompt**

You are a highly skilled bioinformatics assistant. Your task is to generate a structured JSON output that contains code and detailed reasoning to reproduce a specific biological insight.

You are provided with:

1. A **High-Level Insight**, including:
    - A **summary**: a concise statement of a biological finding.
    - A **description**: a more detailed explanation of how the finding was derived — including techniques (*e.g.*, scRNA-seq, UMAP, clustering), key genes, cell types, visualizations, or statistical analyses.

- A **paragraph**: parts of text from the paper that underpin the insight, for context. Use the associated paragraphs to identify any additional details that could help you in retrieving relevant code and creating code snippets.

2. A Python dictionary of **relevant code files**:

- Keys: File paths of scripts identified as potentially relevant. (*e.g.*, '"figures/plot_ETV5.py"')
- Values: Full contents of each source code file.

**Definition of High-Level Insight**: A high-level insight captures a major biological takeaway from the study — the kind you would find in a figure legend, abstract, or conclusion section. These insights typically reflect the biological **what**, **why**, and **how** of a meaningful result.

**Your Task**:

- Analyze the insight carefully to understand the exact biological finding and how it was derived.
- Read and extract ideas from the relevant scripts, identifying any reusable logic, processing steps, parameters, visualizations, or gene/cell type operations.
- Then, **write your own code** in Python language. Your code will operate on a preloaded data object to reproduces this insight. You can name the data object 'adata'.
- Your code should be enclosed using the `<execute>` tag, for example: `<execute> print("Hello World!") </execute>`. **IMPORTANT:** You must end the code block with the `</execute>` tag.
- You can ground your solution in techniques, variable names, or logic from the scripts — but **you must synthesize and write new code that replicates the insight**, not just copy-paste.
- Organize the code into self-contained code blocks, where each block represents a logical step in generating the insight.
- When generating figures or tables, **ensure that the ordering, style, and number of components exactly match those in the original code**.

**Code Output Format**
Each code block must be structured as follows:

- '"code"': Python code needed for a specific step
- '"reasoning"': Explain the purpose of this code step and how it contributes to reproducing the insight
- '"derived_from"': A list of file paths (as strings) where the logic for this step originated or was adapted from

**Important Rules**:

- You may assume the 'adata' object is already loaded in memory.
- You **must not include code for loading 'adata'**.
- Keep the code simple, focused, and biologically relevant.
- Avoid generating fake or overly generic code; always base your logic on the actual insight and provided files.
- For each step, explain both the **"why"** and the **"how"** of the code.
- Ensure each code block does only one logical task (*e.g.*, filtering cells, plotting a violin plot, scoring a gene module).
- **Preprocessing requirement**: Do not introduce any preprocessing steps that are not present in the original source file from which the code is derived.

- **Visualization requirement**: Maintain fidelity to the original visualizations and tables regarding their order, style, and number of components. Match plot parameters, axes, angles, colors, and any labeling conventions precisely.
- **Another visualization requirement**: One figure should have only one plot. If the original code has multiple plots in one figure, split them into separate figures. Show all plots in the original order.
- Do not assume columns are in any fixed order. Instead, locate columns by their exact feature names or headers.

*tree representation of code repository files*

**Final Output Format:**

```json
{
  "Insight Summary": {
    "summary": "...",
    // copied from the insight summary
    "description": "...",
    // copied from the insight summary
    "code_blocks": [
      {
        "code": "<execute> ... </execute>",
        // the code to reproduce the insight
        "reasoning": "...",
        // the reasoning for the code
        "derived_from": ["path/to/file1.py", ...]
        // the files from which this code was derived
      },
      ...
    ]
  }
}
```

## A.2   PROMPTS FOR QUESTION GENERATION

The second stage of the HEUREKABENCH framework consists of converting validated insights into questions. Each insight is represented with three components − *(i) summary*, *(ii) experimental techniques*, and *(iii) grounding text*. An LLM takes the entire information to create two OEQs and two MCQs per insight. Below, we provide the prompts for creating these question types.

---

**OEQ generation prompt**

I am designing assignments for my PhD students on single-cell omics data. The assignment is based on a published scientific article presenting new research findings. I want the PhDs to analyze the data and derive insights similar to those presented in the article, but without access to the article itself. That way, they will have to rely on their analytical skills and understanding of the data rather than simply recalling the article's conclusions or general biological knowledge.

**Assignment structure:**

- My PhD students will receive a dataset from the article, containing single-cell omics data.
- They will *not* have access to the article itself.

---

- They will be required to analyze the data and answer a series of questions that test their ability to interpret patterns and derive biological insights from the dataset.

**Your task:**
- I will provide a list of key insights extracted from the article. Each insight contains a summary, the description of how the insight was derived by the authors, and the associated paragraphs from the paper that support this insight.

- For each insight, create two (2) open-ended questions that assess students' ability to reason through the data to reach similar conclusions. **The questions should together cover different aspects of the insights and its derivation. The questions must remain strictly grounded in the provided insight** without introducing hallucinations.

- **Questions should be mostly designed based on the derivation of the insight** and not be simply factual recall of the insight's summary.

- Use the associated paragraphs to identify any additional details that could help you design more challenging questions.

- Question should not rely on your external knowledge. The desired correct answer(s) must reflect conclusions that can be drawn directly from the omics dataset, not just recall of factual statements.

**Guidelines for creating questions:**
- **In addition to the questions, provide the correct answer(s) for each question**, based strictly on the dataset-derived insight.

- **Please provide the answer(s) immediately after each question.**

- Answers should focus on the findings themselves and not mention the specific methods or tools used to obtain them (*e.g.*, SCENIC, differential gene expression analysis, CellDB).

The goal is to translate each research insight into a data-grounded question that tests PhD students' analytical reasoning and interpretation skills in the context of single-cell omics data.

**Output format:** For each insight, provide the question and the answer in the following format. Do not include any additional text or explanations before or after the output.

**Insight:** [Your insight here]
**Question1:** [Your question here]
**Answer1:** [Answer based on the dataset/insight]
**Question2:** [Your question here]
**Answer2:** [Answer based on the dataset/insight]

**Few-shot examples:**
*[few-shot examples of desired questions and answers form]*

**Examples of what not to do:**
- Do not use double-barreled formulations such as "How does X differ, and what might this suggest. . . ?" or "How do X influence Y, and what is the impact on Z?".

- **Do not combine two sub-questions into one (*e.g.*, "How. . . , and . . . ?")**. Instead, split them into separate, single-focus questions.

- Do not create questions that are rephrasing the summary of the insight. These can usually be answered without the data analysis.

- **Do not create questions that ask which techniques** would a PhD student use to obtain certain result.

- **Do not create questions that ask which techniques** would a PhD student use to obtain certain result.
- Questions should **go beyond simple answers like "increase/decrease" or "yes/no."** They should be open-ended, requiring PhDs to explore different possibilities and justify their reasoning.
- Questions **should not specify the method by which the answer must be obtained**, leaving room for students to choose their own approach.

*[few-shot examples of not-desired questions and answers form]*

**Extracted insights (their summaries and how they were derived):**

**Below is the source material for generating the questions.** Focus more on the derivation of the insights, not just their summaries. The insights are:
*list of insights from the paper*

**Please generate two (2) open-ended questions with their answers for each insight following the above instructions.**

---

**MCQ generation prompt**

I am designing assignments for my PhD students on single-cell omics data. The assignment is based on a published scientific article presenting new research findings. I want the PhDs to analyze the data and derive insights similar to those presented in the article, but without access to the article itself. That way, they will have to rely on their analytical skills and understanding of the data rather than simply recalling the article's conclusions or general biological knowledge.

**Assignment structure:**

- My PhD students will receive a dataset from the article, containing single-cell omics data.
- They will *not* have access to the article itself.
- They will be required to analyze the data and answer a series of questions that test their ability to interpret patterns and derive biological insights from the dataset.

**Your task:**

- I will provide a list of key insights extracted from the article. Each insight contains a summary, the description of how the insight was derived by the authors, and the associated paragraphs from the paper that support this insight.
- For each insight, create two (2) multiple-choice questions that assess students' ability to reason through the data to reach similar conclusions.
    - The questions should together cover different aspects of the insights and its derivation. The questions must remain strictly grounded in the provided insight without introducing hallucinations.
- Questions should be mostly designed based on the derivation of the insight and not be simply factual recall of the insight's summary.
- Use the associated paragraphs to identify any additional details that could help you design more challenging questions, including plausible but tricky wrong answers (hard negatives).
- Question and correct answer should not rely on your external knowledge. The correct answer(s) must reflect conclusions that can be drawn directly from the omics dataset, not just recall of factual statements.

- Tips for designing hard questions with hard negatives:
  - Wrong answers should simulate realistic misinterpretations of the data, premature conclusions, or confusions between similar cell types/genes/pathways. These are more cognitively demanding for PhD students to distinguish.
  - Avoid irrelevant or obviously false options. Each incorrect option should reflect a misguided but well-intentioned line of reasoning from someone analyzing the dataset.
- Questions can be either:
  - *Single-answer*: one correct option (*e.g.*, "D")
  - *Multiple-answer*: more than one correct option (*e.g.*, "A,C,D")

**Guidelines:**
- Randomize the position of the correct answer(s) among the options.
- Avoid phrasing that suggests PhD students need to recall the article or authors' conclusions. Use neutral language focused on data interpretation, such as "the data indicate," "analysis of the dataset suggests," or "based on gene expression patterns."
- The question should not specify the exact methods to use to derive the answer from the data. The PhD students should be able to determine the appropriate analysis methods based on the question and their understanding of single-cell omics.
- The questions should be as open-ended as possible, allowing PhD students to explore the data and derive their own conclusions. Do not specify the exact analysis methods or outcomes in the questions.
- **If for the answer, there are multiple correct options (*e.g.*, cells, genes, etc.), the answer should be splitted in multiple options**, *e.g.*, A) cell type 1, B) cell type 2, C) cell type 3, D) cell type 4. **The correct answer should be a combination of these options**, *e.g.*, "A,C" or "B,D".
- **Please provide many questions that cannot be answered without the data.** This kind of data can be sometimes found in the description of the insight's derivation.

The goal is to translate each research insight into a data-grounded question that tests PhD students' analytical reasoning and interpretation skills in the context of single-cell omics data.

**Output format:** For each insight, provide the question and answer options in the following format. Do not include any additional text or explanations before or after the output.

**Insight:** [Your insight here]
**Question1:** [Your question here]
A) [Option A]
B) [Option B]
C) [Option C]
D) [Option D]
**Answer1:** [Correct option(s) here, *e.g.*, "A,C"]
**Question1:** [Your question here]
A) [Option A]
B) [Option B]
C) [Option C]
D) [Option D]
**Answer2:** [Correct option(s) here, *e.g.*, "A,C"]

**Few-shot examples:**

*[examples of desired question and answers form]*

**Examples of what not to do:**

> - Do not create questions that are rephrasing the summary of the insight. These can usually be answered without the data analysis.
> - Do not create questions that ask which techniques would a PhD student should use to obtain certain result.
>
> *[examples of non-desired question and answers form]*
>
> **Extracted insights (their summaries and how they were derived): Below is the source material for generating the questions.** Focus more on the derivation of the insights and the associated paragraphs from the paper, not just the insight summaries. The insights are: *list of insights*
>
> **Please generate two (2) multiple-choice questions for each insight following the above instructions.**

### A.3 ADAPTATION OF HEUREKABENCH FOR OTHER SCIENTIFIC DOMAINS

HEUREKABENCH is a general framework that can be used to instantiate benchmarks for evaluating AI co-scientists in other scientific domains. Below, we provide key changes to adapt to another domain D:

- The framework consists of two stages, as shown in Figure 1, namely, *insight generation* and *question generation*, which remain unchanged.

- The automated components within each stage comprise different LLM-based modules (*e.g.*, *InsightExtractor*, *CodeDescriber*, *CodeMatcher*, *CodeGenerator*) require minor edits in their prompts (as shared in Section A.1 and A.2) during this adaptation.

- To adapt the prompts, the required changes include the domain names and a few-shot examples. Such examples can be generated by running the modules without any examples first, while all other instructions are present. Then, some outputs can be selected to include in the prompt for the final execution. Note that these modules can be run without examples; however, in practice, since LLMs are the core of these modules, it is recommended to provide a few-shot in-domain examples.

- Finally, the framework needs human experts with enough experience working in the domain to understand and analyze the outputs of automated modules. As a potential future work, automating this step reliably would convert the semi-automated HEUREKABENCH into an automated framework.

The above steps make minimal changes to the HEUREKABENCH, enabling straightforward adaptation of the framework to new domains.

### A.4 ADDITIONAL DETAILS ON QUESTION FILTERING

Our question generation pipeline converts validated insights into questions. However, since we utilize LLMs for this task, it remains essential to perform additional filtering to develop a more robust benchmark. For the first **automatic filtering** stage, we rely on closed-source LLMs. We use both GPT-4o and Claude-4-Sonnet to answer all OEQs and MCQs. For MCQs, we discard any questions that both GPT-4o and Claude-4-Sonnet correctly answered. For OEQs, we use G-Eval (Liu et al., 2023) to assign a score between 1 and 5, and eliminate questions that received scores above 3.0 for both LLMs. This aims to reduce the risk of LLM-based agents answering from pre-training knowledge. The second **manual filtering** stage first checks for potential hallucinations and duplication by cross-referencing with the insights representation. Next, for insights containing multiple findings but validation only for a subset (*e.g.*, if the *experimental techniques* suggest *immunofluorescence staining* was used by authors to validate some part of the insight, we mark that part as not validated), we remove questions derived from the non-validated components.

## B    Details on Evaluation in HeurekaBench framework

One of our key contributions is the adaptation of G-Eval (Liu et al., 2023) to evaluate data-driven agent responses against ground-truth conclusions derived from scientific findings, while simultaneously penalizing responses that rely predominantly on pre-training knowledge of the LLM rather than exploring the provided dataset. Moreover, the generated response may itself contain novel findings, an essential aspect of the AI co-scientist, when it is tasked with discovering patterns from experimental data. To support this, we design a grading rubric grounded in *atomic facts* extracted from free-form text. We define atomic facts as minimal, verifiable units, such as conditions, trends, and conclusions, that can be systematically compared across texts. Concretely, we instruct the GPT-4o grader to first decompose responses into atomic facts, and then assess whether each ground-truth fact is fully present, partially present, or absent in the agent's answer. The score increases with the extent and number of correctly captured atomic facts, provided that no statement contradicts the ground truth. The full evaluation prompt is included below.

---

**G-Eval system prompt for grading sc-HeurekaBench (OEQs)**

You are a full professor in single-cell biology evaluating your PhD student's responses to questions.

---

**G-EVAL prompt grading sc-HeurekaBench (OEQs)**

You are grading a PhD student's answer to an open-ended research question about a single-cell biology dataset. The task requires analysis of the dataset to derive the answer to the question. You will be given the student's answer and the ground-truth (GT) answer.

**Evaluation Steps:**

1. Extract atomic facts from the GT and label them F1..Fn. An atomic fact is a claim that can be objectively verified (including but not limited to: cell type/condition, direction/magnitude of change, gene/pathway names, statistical evidence, method-/application, conclusion).

2. For each Fi, classify the student's coverage as:

   - **PRESENT:** fact included with the same meaning AND explicitly tied to dataset-derived quantitative/statistical outputs or cluster/subtype identifiers. PRESENT requires evidence that the student directly engaged with the dataset-defined labels.
   - **PARTIAL:** This group includes:
     - Facts with the correct meaning but supported only by descriptive biology or lists of plausible options (*e.g.*, cell type names, marker/pathway lists), without dataset-level numbers or identifiers.
     - Facts whose support is vague or hedged, using terms such as "*e.g.*," "seems," "maybe," "likely," "generally," "typically," "usually," "for example," or similar language.
     - Facts that appear to rely solely on general biological knowledge or recall, without direct reference to the dataset.
     - Answers that list multiple plausible options (*e.g.*, marker genes, pathways, or cell types) without presenting dataset-based evidence.
   - **MISSING:** fact is not mentioned.
   - **INCORRECT:** wrong fact or fact that is contradictory to some GT fact.

3. Identify contradictions: any statement that is in direct conflict with the GT (including but not limited to: opposite direction, wrong cell type/condition).

   - Contradictions should only be judged relative to GT, not relative to biological knowledge outside GT.
   - Facts marked as MISSING do not count into contradictions.

- Omitting details, partial coverage, or failing to mention context is not a contradiction. It may affect coverage (PARTIAL/MISSING) but does not count as contradictory.

4. Count number of GT facts that are PRESENT, PARTIAL, MISSING, or INCORRECT. Evaluate only against parts of the answer that address the GT facts.

5. Review the coverage labels for all GT facts and determine the overall correctness score (1–5) based on the scoring rubric below.

6. Presence of additional information that is not part of GT facts, but does not contradict GT, **does not affect the score**. Calculate the score solely based on the GT facts and their coverage labels.

**Scoring (Correctness 1–5, integers only):**

- **5:** All GT facts are marked as PRESENT. No facts are marked as MISSING or INCORRECT. No contradictions of GT facts.

- **4:** Most or all GT facts are marked as PRESENT. Detailed, dataset-grounded analysis is valid even if it does not repeat the GT fact's broader/general terms, as long as it conveys the same underlying fact. Facts labeled as MISSING are allowed. Facts labeled as INCORRECT are not allowed.

- **3:** Some GT facts (but not all) are marked as PRESENT. At least one GT fact is marked as PARTIAL or MISSING. Minor contradictions of GT facts are allowed. Facts labeled as MISSING are allowed.

- **2:** No GT facts are marked as PRESENT; some are PARTIAL. Student includes broad or generic options (*e.g.*, list of plausible genes with no explanations), which seems like recall of biological knowledge and not dataset evidence. Minor contradictions of GT facts are allowed.

- **1:** All GT facts are marked as MISSING; most GT facts are marked INCORRECT; there are major contradictions of GT; the student explicitly states inability to answer / no data available.

**General Rules:**

- Grade only against the GT answer; ignore outside knowledge.

- Focus only on parts addressing GT facts; ignore unrelated detail unless contradictory.

- It does not matter if GT facts are not emphasized or the main focus – only matters whether they are present in the student's answer and not vague.

- Style, length, or phrasing differences (including paraphrases/synonyms) do not matter as long as GT facts are covered with dataset grounding.

- Dataset grounding (required for PRESENT): explicit quantitative/statistical evidence or dataset identifiers (*e.g.*, percentages, fold changes, p-values, cluster IDs, enrichment scores).

- Assess additional information, not related to GT fact: penalize only if the additional information is contradictory to the GT.

- Extra details related to the GT fact (*e.g.*, subtypes, fold changes, statistics) are valid if they are dataset-based and consistent with the GT. Such details should not "obscure" correctness score as long as they are neither contradictory nor vague. However, vague or unsupported details—especially those resembling general biological recall (*e.g.*, lists of plausible genes, cell types, or pathways)—should downgrade the related GT fact from PRESENT to PARTIAL.

- If details tied to the GT fact rely on vague or hedging language (*e.g.*, "*e.g.*," "likely," "for example,"), or involve listing many plausible options (*e.g.*, genes, pathways, cell types), the related GT fact should likewise be downgraded from PRESENT to PARTIAL.

> - If the GT fact appears only as part of an extensive list (including but not limited to: lists of plausible genes, pathways, or cell types), without dataset-specific grounding, the answer should be downgraded from PRESENT to PARTIAL. This applies even if the GT fact is technically included, since its support is obscured by recall-like listing rather than dataset evidence.
> - All GT facts are treated equally. No fact is inherently more important than others.
>
> **Provided Answer:**
>
> `{answer}`
>
> **Ground Truth Answer (GT):**
>
> `{gt_answer}`
>
> **Output Format:**
> - Output the numerical rating wrapped in `<rating></rating>` tags.
> - Do not include extra text outside these tags.
> - Example:
>
> `<rating>3</rating>`
>
> **Your Response:**

## C DETAILS ON SC-HEUREKABENCH BENCHMARK

### C.1 SELECTED PUBLICATIONS FOR INSIGHT GENERATION

Below, we provide the 13 publications which were used to create the sc-HEUREKABENCH benchmark. All of these have at least one validated insight and hence constitute the validated papers. Additionally, we mention the number of OEQs and MCQs created from each publication after the question generation stage of the HEUREKABENCH framework.

Table 5: List of papers used for creating sc-HEUREKABENCH benchmark. Reported are the OEQs and MCQs for each publication, for a total of 50 OEQs and 50 MCQs in sc-HEUREKABENCH.

| Authors | Journal | #OEQs | #MCQs |
|---|---|---|---|
| Lazarescu et al. (2025) | Nature Genetics | 4 | 4 |
| Yu et al. (2025) | Nature Genetics | 8 | 8 |
| Maatz et al. (2025) | Nature Cardiovascular Research | 1 | 1 |
| Wang et al. (2025) | Nature | 0 | 1 |
| Zhang et al. (2024) | Nature | 2 | 3 |
| Gu et al. (2024b) | Nature | 2 | 2 |
| Kedlian et al. (2024) | Nature Aging | 11 | 10 |
| Zwick et al. (2024) | Nature Cell Biology | 2 | 2 |
| Isola et al. (2024) | Nature Aging | 6 | 5 |
| Rexach et al. (2024) | Cell | 3 | 3 |
| Hoo et al. (2024) | Cell Systems | 5 | 7 |
| Escoubas et al. (2024) | Cell | 2 | 1 |
| Li et al. (2024) | Cell Stem Cell | 4 | 3 |
| Total | | 50 | 50 |

## C.2  ADDITIONAL DETAILS ON PUBLICATIONS

In this section, we highlight some key features of each publication to demonstrate the diversity within sc-HEUREKABENCH. In particular, we summarize the location (*e.g.*, tissues, organs) and cell conditions (*e.g.*, disease, treatments) whenever available. Lazarescu et al. (2025) sequence and characterize cells from human subcutaneous or visceral adipose tissues. Yu et al. (2025) focuses on neuroblasts from patients with high-risk neuroblastoma (*i.e.*, cancer) before and after chemotherapy. Maatz et al. (2025) study patients with myocarditis (*i.e.*, inflammation of the heart), in particular, the left ventricular endomyocardium in relation to COVID-19 infection and vaccination. Wang et al. (2025) focus on the prefrontal cortex and the primary visual cortex (*i.e.*, brain) across five developmental stages, from the first trimester to adolescence. Zhang et al. (2024) studies human embryonic limb development. Gu et al. (2024b) sequences cells from the intestinal immune system, specifically, activation of $T_{reg}$ cells in inflammatory conditions. Kedlian et al. (2024) creates an atlas to study the aging process in the intercostal muscle in humans. Zwick et al. (2024) study the nutrient absorption at different segments of the small intestine in mice and humans. Isola et al. (2024) is another atlas study on aging in mouse ovaries. Rexach et al. (2024) sequence cells from human brains under three dementia related diseases. Hoo et al. (2024) focuses on the effect of three pathogens, namely, *P. falciparum*, *L. monocytogenes*, and *T. gondii* on the placenta. Escoubas et al. (2024) studies microglia cells (*i.e.*, macrophages in the brain). Li et al. (2024) analyzes the interaction between trophoblast and uterine natural killer (uNK) cells (*i.e.*, uterine mucosa) to understand how uNK cells affect placentation.

## C.3  CATEGORIZATION OF OPEN-ENDED QUESTIONS (OEQS)

In this section, we classify OEQs into the following categories:
*(i) heterogeneity analysis* deals with cell compositions or characterization in a fixed cell condition (*e.g.*, "How many distinct transcriptomic states can be identified in neuroblastoma neoplastic cells, and how are these states characterized?"),
*(ii) condition/treatment analysis* focuses on a certain external or observation condition which is varied (*e.g.*, "What changes can be observed in cell state proportions before and after chemotherapy in high-risk neuroblastoma patients?"),
*(iii) pathway analysis* concentrates on pathway-specific questions (*e.g.*, "What inflammasome activation pathways are identified as increased in Post-Vaccination myocarditis?"),
*(iv) key gene analysis* asks about particular gene or transcriptional changes (*e.g.*, "What characteristic genes define the novel neuromuscular junction (NMJ) accessory population identified in the study?"),
*(v) cellular functioning analysis* queries on cell level shifts, particularly related to their behaviour (*e.g.*, "How do Hofbauer cells (HBCs) demonstrate plasticity in their response to different pathogens based on the dataset?"),
*(vi) cell-cell communication analysis* is specific to the final publication (Li et al., 2024), which deals with cell-cell communication between uNK and trophoblast cells (*e.g.*, "How do uterine natural killer (uNK) cell-derived cytokines influence the expression of cytokine receptors in extravillous trophoblasts (EVTs)?").

It is important to note that, although we attempted to create distinct categories, questions can require agents to perform multiple analyses to generate an accurate answer. We provide the number of OEQs per category for sc-HEUREKABENCH and sc-HEUREKABENCH-Lite in Figure 4, which shows that the distributions of tasks across both versions are similar.

## C.4  INVALIDATED INSIGHTS

In this section, we discuss design choices related to insight validation during the creation of the sc-HEUREKABENCH. We perform manual validation of workflows generated by *CodeGenerator* corresponding to an insight. Note that *CodeGenerator* is instructed to ground its outputs in files from open-source code repositories. Consequently, it constructs multi-step workflows and references the subset of files used to generate each step. During this validation, we always perform three minor code edits to ensure a higher chance of successful execution: *(i)* load appropriate dataset(s) because *CodeGenerator* is instructed to assume a pre-loaded data object, *(ii)* map stable Ensembl

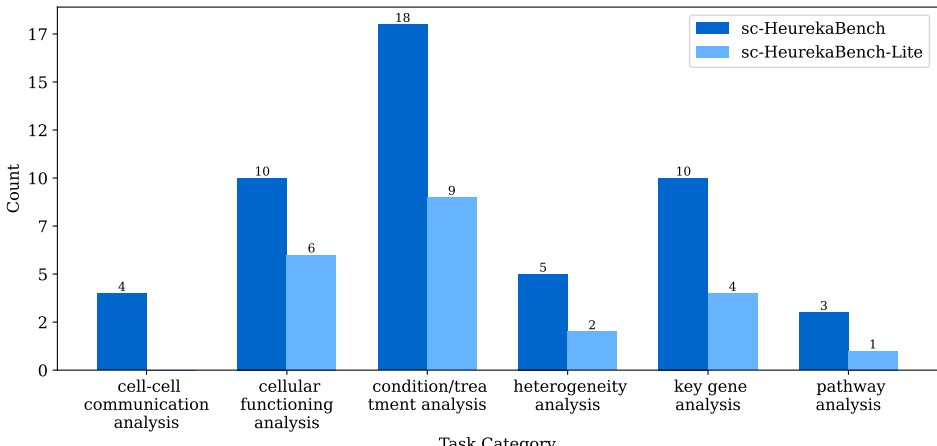

Figure 4: **Question distribution per task category.** There are six different categories, with the distribution across both versions of the benchmark following a similar distribution. The questions related to cell-cell communication analysis are obtained only from Li et al. (2024), which is not part of the sc-HEUREKABENCH-Lite.

gene identifiers[2] present in the experimental dataset to their common names generally found in the insights, and *(iii)* change the variables and metadata names as per the experimental dataset.

As a result of this validation step, some insights are invalidated. The primary reasons include: *(i)* **insufficient information about the dataset used to derive the insight**. In such cases, running the workflow on an alternative dataset produces results that differ from those reported in the paper. For example, the workflow specifies bulk RNA-seq data, while only scRNA-seq data is available from CellxGene (CZI Cell Science Program et al., 2025). *(ii)* **inconsistencies between the workflow's requirements and the available dataset**. For instance, a workflow may reference sub-cluster information that the dataset does not provide. *(iii)* **overly generic insights** that cannot lead to meaningful questions (*e.g.*, *The study presents a comprehensive human skeletal muscle aging atlas, providing a valuable resource for studying muscle aging across species.*). While we mark the above insights as invalid in our benchmark, we expect that these could potentially be validated if the correct set of input information is provided.

### C.5 Comparative Analysis with BaisBench

In this section, we perform a comparative analysis with BaisBench (Luo et al., 2025). BaisBench generates only multiple-choice questions (MCQs) from scientific publications using a single LLM, which (i) do not undergo any validation, and (ii) do not use the corresponding experimental data or code repositories for grounding the questions. In contrast, our proposed HEUREKABENCH utilizes scientific publications along with their corresponding code repositories to ensure that each question-answer pair (i) depends on manually validated insights through code execution, and (ii) is grounded in corresponding experimental data.

Furthermore, we perform quantitative analysis to compare the quality of the generated questions. We use GPT-5 (OpenAI, 2025) to answer MCQs from both datasets. In this setup, the LLM is given access only to questions, not to any experimental data. Hence, it tests the data-driven nature of the questions, *i.e.*, whether they can be answered correctly based on knowledge recall. GPT-5 answered 53.37% of questions accurately on BaisBench, compared to 34.69% on sc-HEUREKABENCH, indicating that our framework yields questions that are more difficult to answer with recall alone.

---

[2] https://www.ensembl.org/index.html

# D DETAILS ON EXPERIMENTS

## D.1 SCRAPING PUBLICATIONS AND EXPERT INSIGHTS FROM FLYBASE

To curate pairs of expert insights and publications from FlyBase (Öztürk-Çolak et al., 2024), we focus on the "Other Comments" section of gene-level report pages. We select the following gene identifiers: `FBgn0000490`, `FBgn0001077`, `FBgn0001139`, `FBgn0001168`, `FBgn0001180`, `FBgn0003448`, `FBgn0003731`, `FBgn0003996`, `FBgn0004644`, `FBgn0004647`. We collect five random expert insights with open-access publications per identifier, resulting in a total of 50 pairs. Within each insight, we append the abbreviated gene names with their full names from the gene report pages for additional context to LLM-Judge.

## D.2 GPT-4O JUDGE FOR MATCHING OUR AND EXPERT INSIGHTS

To validate our *InsightExtractor*, we rely on G-Eval (Liu et al., 2023) to compare if our generated insights from the publication resemble the expert-annotated findings from the corresponding dataset. We instruct the G-Eval to label relatedness as *strongly related*, *weakly related*, or *unrelated*. The entire prompt is available below.

---

**G-EVAL system prompt for InsightExtractor validation**

You are an expert evaluating conceptual alignment between AI-generated insights and a scientist-derived insight.

---

**G-EVAL prompt for InsightExtractor validation**

You will be given a list of LLM-generated insights and a single scientist-derived insight. Your task is to assign a single score according to the following rules.

**Scoring (Relatedness 1–3, integers only):**

- **3 (strongly related):** At least one LLM insight is strongly related to the scientist-derived insight.
- **2 (weakly related):** No strongly related insights, but at least one is related.
- **1 (unrelated):** All LLM insights are unrelated.

**LLM-derived insights:**
*[list of llm insights]*

**Scientist-derived insight:**
*[list of insight derived by scientists]*

**Output Format:**

- Output the numerical rating wrapped in `<rating></rating>` tags.
- After the rating, output an explanation wrapped in `<explanation></explanation>` tags.
- Do not include extra text outside these tags.
- Example:

`<rating>2</rating>`

---

## D.3 MANUAL ANALYSIS OF AI CO-SCIENTIST ANSWERS

One of the important reasons for designing sc-HEUREKABENCH is to understand the failure modes of current LLMs used as AI co-scientists. We manually analyze agent responses to provide preliminary insights below:

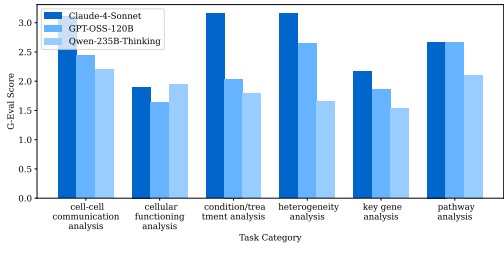
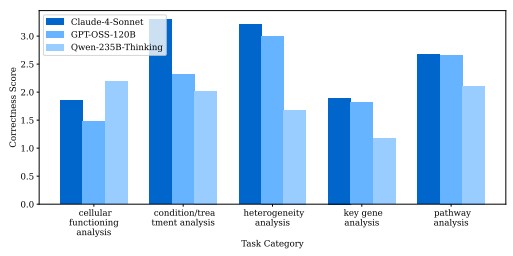

(a) sc-HEUREKABENCH                (b) sc-HEUREKABENCH-Lite

Figure 5: **Score distribution per task category.** Both versions of the benchmark exhibit a similar trend in score distributions across three evaluated LLMs, with the closed-source model outperforming the open-source model in all categories. Cell–cell communication category is sourced exclusively from Li et al. (2024), which is not included in sc-HEUREKABENCH-Lite.

- **incorrect scientific skills**, where the agent recalls scientific knowledge from pre-training, *e.g.*, instead of calling a tool for gene set enrichment (pathway) analysis, it recalls a random number of canonical gene markers for specific pathways (or from the MCQ options) and identifies their mean expression to suggest if a pathway is activated. This is not a scientifically correct way to perform pathway analysis, especially within the Biomni-E1 environment, which provides appropriate tools (*e.g.*, gene_set_enrichment_analysis). Furthermore, in several cases, the agent does not explore all the metadata columns and thus cannot find the suitable information.

- **lack of environment exploration**, where, although the agent uses only a few of the retrieved tools, it doesn't explore and consider additional components of the environment that can enable a more informed and holistic response

- **hallucinations**, where the agent directly generates an answer without any analysis or incomplete analysis, resulting in a response that lacks meaningful data-driven conclusions.

- **other**, includes the agent (i) writing large code blocks instead of small, step-wise code snippets, (ii) being unable to take code execution errors into account, and (iii) relying on known literature to eliminate options from MCQs (which actively goes against the idea of data-driven discovery)

These failure behaviours are more prevalent in open-source LLMs than closed-source variants. Equipping open-source LLMs with these skills through appropriate post-training approaches can significantly enhance their adoption as agents for scientific discovery.

Furthermore, we share the correctness scores of Claude-4-Sonnet, GPT-OSS-120B, and Qwen3-235B-THINKING LLMs within the Biomni-E1 environment for sc-HEUREKABENCH and sc-HEUREKABENCH-Lite in Figure 5. We observe that the closed-source LLM consistently outperforms its open-source counterparts, particularly in condition/treatment analysis, heterogeneity analysis, and cell-cell communication analysis, with Claude-4-Sonnet achieving performance up to $\sim 2\times$ that of open-source LLMs. The more challenging categories for all LLMs include key gene analysis and cellular func-

Table 6: Detailed inter-rater agreement metrics for two closed-source judges with GPT-4o.

| LLM Judges | Spearman's rank correlation | Cohen's kappa $\kappa$ |
|---|---|---|
| **Claude-4-Sonnet planner** | | |
| Claude-4.5-Sonnet | 0.863 | 0.778 |
| Gemini-2.5-Pro | 0.800 | 0.651 |
| **GPT-OSS-120B planner** | | |
| Claude-4.5-Sonnet | 0.859 | 0.849 |
| Gemini-2.5-Pro | 0.798 | 0.746 |
| **Qwen3-235B-THINKING planner** | | |
| Claude-4.5-Sonnet | 0.793 | 0.789 |
| Gemini-2.5-Pro | 0.766 | 0.726 |

tioning analysis, where Claude-4-Sonnet achieves 1.90 and 2.16 correctness scores, respectively, while other LLMs achieve lower scores, as low as 1.53 in key gene analysis by Qwen3-235B-THINKING. Further, we observe that all models exhibit a similar trend in correctness scores across task categories in both versions of the benchmarks.

### D.4 INTER-RATER AGREEMENT STATISTICS

As detailed in Section 4.3.1, we perform inter-rater agreement studies between three closed-source LLM judges, GPT-4o, Claude-4.5-Sonnet (Anthropic, 2025b), and Gemini-2.5-Pro (Comanici et al., 2025). We computed Spearman's rank correlation and the unweighted Cohen's kappa ($\kappa$) with a quadratic penalty between the correctness scores assigned by GPT-4o and those assigned by other judges. We report these two statistics for three planner LLMs (Qwen3-235B-THINKING, GPT-OSS-120B, Claude-4-Sonnet) within the Biomni agent in Table 6.

## E EXAMPLES FROM SC-HEUREKABENCH

We provide two sample examples for OEQs and MCQs from the sc-HEUREKABENCH benchmark below.

### E.1 OPEN-ENDED QUESTIONS (OEQS)

---

**Example of questions from sc-HEUREKABENCH OEQ**

**What changes in cytokine expression are observed in the aging muscle microenvironment?**

Aging muscle exhibits increased expression of the pro-inflammatory cytokine IL6 within multiple vascular (SMCs, pericytes and a trend in arterial endothelial cells) and stromal cells (tenocytes and fibroblasts) and decreased expression of IGF1 in fibroblasts.

**What changes can be observed in cell state proportions before and after chemotherapy in high-risk neuroblastoma patients?**

The proportions for ADRN (adrenergic)-baseline and ADRN-proliferating populations decreased after therapy. Conversely, the ADRN-calcium, ADRN-dopaminergic, and Interm-OXPHOS populations exhibited significant increases after therapy. Mesenchymal neoplastic cells (MES) cells made up ¡10% of neoplastic cells in most samples, but some samples contained a high frequency of MES cells with significant post-therapy changes.

---

### E.2 MULTIPLE-CHOICE QUESTIONS (MCQS)

---

**Example of questions from sc-HEUREKABENCH MCQ**

**Which macrophage subset is observed to reduce after therapy according to the dataset?**
- (A) IL18+ macrophages
- (B) THY1+ macrophages
- (C) Proliferating macrophages
- (D) VCAN+ macrophages

**Analysis of the dataset suggests an increase in which cell types in aged human skeletal muscle?**
- (A) B cells
- (B) Schwann cells
- (C) T cells
- (D) Vascular cells

---

## F EXAMPLES FROM SC-HEUREKABENCH-TU

We provide two sample examples from the sc-HEUREKABENCH-TU benchmark below. Note that this benchmark contains OEQs only.

---

**Example of questions from sc-HEUREKABENCH TU**

**How does the expression of collagen and collagenase pathways change in fibroblast-like stromal cells with age?**

The expression of collagen remains unchanged with age in fibroblast-like stromal cells, but the collagenase pathway is downregulated.

**What stage-specific ligand-receptor interactions can be identified in the developing limb from the dataset?**

The dataset identifies stage-specific ligand-receptor interactions, such as WNT5A-FZD10 in distal mesenchyme, JAG1-NOTCH1 in posterior mesenchyme, and FGF8-FGF10 in ectoderm and mesenchyme.

---

## G TASK PROMPTS FOR AGENTS

In this section, we share the task prompts provided to the agents to answer either OEQs or MCQs.

---

**Task prompt to agents for sc-HEUREKABENCH (OEQs)**

Task: Analyze the provided single-cell dataset and answer the biology question.

Input Data:
{data_paths}

Question:
{question}

Output Format:
Return the summary of an answer wrapped inside XML-style tags <solution> and </solution>.

Guidelines for the output format:

- Base the answer strictly on the results derived from the dataset.
- Provide a fact-based summary (not a narrative or manuscript-style report).
- Do not use extra formatting such as bullet points or section headers.
- Include all key findings that directly address the question, emphasizing those most relevant to the answer.

---

**Task prompt to agents for sc-HEUREKABENCH (MCQs)**

Task: Analyze the provided single-cell dataset and answer the biology question by selecting all correct options.

Input Data:
{data_paths}

Question:
{question}

---

Answer Choices:
```
<answer choices>
```

Output Format:
Return the selected options as a comma-separated list of letters wrapped inside XML-style tags `<solution>` and `</solution>`.
For example: `<solution>A,C</solution>`

## H COMPARISON OF BASELINE LLM WITH LLM-BASED AGENT

To quantify the benefit of using agents, we compare a baseline LLM (Claude-4-Sonnet) with Biomni using the same model. The results, presented in Table 7, show that incorporating the agent substantially improves performance on both OEQs and MCQs. Notably, even one of the current top-performing LLMs scores below 2 on the OEQs and performs worse than most open-source models tested with Biomni on MCQs.

Table 7: Comparision of baseline and agentic performance on sc-HEUREKABENCH with Claude-4-Sonnet. Accuracy, recall, and precision metrics are denoted in %. Higher metric values are better.

| | OEQs | MCQs | | |
| --- | --- | --- | --- | --- |
| Model | Correctness [1-5] | Accuracy | Recall | Precision |
| Claude-4-Sonnet | 1.90 | 22.00 | 65.50 | 45.50 |
| Biomni with Claude-4-Sonnet | 2.56 | 44.00 | 85.00 | 66.33 |

## I THE USE OF LARGE LANGUAGE MODEL (LLMS)

In our work, we have utilized LLMs as a general-purpose assistant tool to enhance the clarity of writing, phrasing, and grammar. LLMs were not used for research ideation, experimental design, or result interpretation. All ideas, experiments, and analyses were carried out by the authors. However, LLMs were utilized for various modules within our proposed framework and as part of LLM-based agents, as documented in the previous sections and the main text.

