# OpenReview forum: "HeurekaBench: A Benchmarking Framework for AI Co-scientist"
_ICLR.cc/2026/Conference — ICLR 2026 Poster_

### Official Review · Reviewer_eTC4 · 2025-10-31

**Soundness:** 2
**Presentation:** 3
**Contribution:** 2
**Rating:** 4
**Confidence:** 3

**Summary:**

This paper introduces HeurekaBench, a pipeline for evaluating large language model (LLM)-based co-scientist agents with open-ended data-driven questions. The pipeline extracts insights matched with code from published studies with code, generate questions and MCQs from these insights. Using this pipeline, the authors create sc-HEUREKABENCH, a benchmark with 50 open-ended questions and 50 MCQs across 41 validated insights.

Success on the benchmark involves the agent deriving conclusions from the question and data that matches the ground-truth insights. They also propose a structured evaluation scheme using an LLM-as-a-judge (G-Eval with GPT-4o) that decomposes responses into atomic scientific facts.

**Strengths:**

S1. Evaluating co-scientist agents is a challenging problem especially for end-to-end analyses. The idea to leverage matched insights/conclusions as a measure is a smart way to evaluate co-scientist agents.

S2. The authors do a decent job validating/ensuring the quality of the dataset. This includes defining what a good agent output is and including a reasonable semi-automatic pipeline to validate code, reproducibility of insights, and questions.

S3. The implementation in a concrete scientific field (single-cell biology) adds a good proof of concept.

S4. The paper is well written with good details in the appendix.

**Weaknesses:**

W1. I struggle with the claim of HeurekaBench as a framework. If the contribution is a framework then I would expect it to be validated in more then one domain to show how this process generalizes. Given the domain expertise required to validate code, insights, and questions in different domains, it seems like the requirements for human validation would be different. This can be further affected by the availability of code and data in different fields. Furthermore, the prompts shared are also customized to the domain of single-cell biology. Thus, if someone were to apply this framework, it does not seem to be a trivial process and essentially requires coming up with a pipeline from scratch.


While I appreciate the authors validating their dataset, the InsightExtractor having 14 weakly or unrelated insights out of 30 and the CodeMatcher matching only 75% of relevant files is worrying given the high bar for quality for an evaluation dataset. At the very least, this again means significant human effort/expertise is necessary to validate and so makes it harder to view the framework as generalizeable.

W2. The paper does not report error bars or statistical significance. Do these results change given different random seeds. Given the understandably significant effort in constructing such a dataset, I worry the number of tasks (and unique publications) is too small to show meaningful/informative performance differences.

W3. Since the evaluation hinges on LLM as a judge, it would be important to know how the LLM scores compare to human expert judgement. We do not know this right now.

**Questions:**

What are steps to ensure that the LLM/agent is doing the correct steps to obtain the correct insights? How do we avoid a hallucinated answer that happens to be correct?

How do we ensure that the LLM has not seen the data/publications in training, especially as new models come out? How does this impact our interpretation of results.

While the approach is different, the the focus on open-ended questions is not new [1]. It would be helpful to contextualize the motivation in the related work.

[1] Gu et al. “BLADE: Benchmarking Language Model Agents for Data-Driven Science.” arXiv preprint arXiv:2408.09667 (2024).

---

> ### Author Response · Authors · 2025-11-21
> **Response to Reviewer eTC4 (1/3)**
>
> We thank the reviewer for their valuable feedback on our submission. We are delighted that the reviewer finds our idea of generating questions from matched insights and conclusions in scientific publications to be a smart way to evaluate co-scientist agents in end-to-end analyses. We appreciate the reviewer’s positive remarks on our efforts to validate and ensure the quality of the benchmark and on instantiating our framework in single-cell biology as a good proof of concept. Finally, we are pleased with the reviewer’s appreciation of our writing and the details in the appendix. Below, we address the reviewer’s questions (Weaknesses and Questions are abbreviated as W and Q):
>
> > W1.1. I struggle with the claim of HeurekaBench as a framework. If the contribution is a framework then I would expect it to be validated in more then one domain to show how this process generalizes. …Thus, if someone were to apply this framework, it does not seem to be a trivial process and essentially requires coming up with a pipeline from scratch.
>
> We appreciate the reviewer’s comments. Our framework comprises a general pipeline that converts scientific publications in a domain into question-answer pairs to test agents for data-driven scientific discovery.  First, we would like to highlight that adapting our framework to other scientific areas requires only domain experts and minimal modifications to the LLM prompts (e.g., the domain name and optional few-shot in-domain examples), with the remaining general instructions unchanged. Second, the main bottleneck to application in another domain lies in human validation, which is the most demanding aspect of all human-validated benchmarks. In response to this question and to make adoption in other domains easier, we now provide concrete steps for instantiating HeurekaBench in Appendix A.3 and highlight the changes from our instantiation of scHeurekaBench.
>
> > W.1.2. While I appreciate the authors validating their dataset, the InsightExtractor having 14 weakly or unrelated insights out of 30 and the CodeMatcher matching only 75% of relevant files is worrying given the high bar for quality for an evaluation dataset. At the very least, this again means significant human effort/expertise is necessary to validate and so makes it harder to view the framework as generalizeable.
>
> We thank the reviewer for their appreciation of our efforts to validate the benchmark's components. First, we would like to draw attention to the performance of InsightExtractor on papers and findings pairs from the Flybase repository, where it extracted insights that were 44 strongly related, 2 weakly related, and only 4 unrelated. Next, we carefully examined the BixBench hypotheses and found that some were designed to be false (i.e., negative of the findings reported in the corresponding paper). After removing those, 21 hypotheses remain: 14 are strongly related, 4 are weakly related, and only 3 are not related. This demonstrates strong insight extraction capabilities. We have now updated the BixBench-related numbers in Section 4.1 and Figure 2 (a) and we are grateful to the reviewer for this very insightful comment.
>
> Next, in case of incorrect code block retrieval, the ground truth and retrieved code blocks contain only minor differences which can be fixed with minor edits during validation (i.e., falls under ‘change the variables and metadata names as per the experimental dataset’ that can be easily checked as mentioned in Appendix C.4). Although our benchmark relies on manual verification for grounding the questions in scientific literature and experimental datasets, unlike other works on creating data-driven benchmarks [1, 2], in our framework, human experts do not need to create tasks, code, and validation from scratch for different datasets.

---

> ### Author Response · Authors · 2025-11-21
> **Response to Reviewer eTC4 (2/3)**
>
> > W2. The paper does not report error bars or statistical significance. Do these results change given different random seeds. Given the understandably significant effort in constructing such a dataset, I worry the number of tasks (and unique publications) is too small to show meaningful/informative performance differences.
>
> We appreciate the reviewer’s suggestion to report the averaged performance across multiple runs. We have now completed the Biomni agents three times to demonstrate robustness across runs. In particular, we selected the Qwen3 family of models, GPT-OSS-120B, and Claude-4-Sonnet to demonstrate that the overall rankings and conclusions remain consistent with our initial results. The correctness scores of the models are reported below with mean and standard deviation:
>
> Qwen3-32B: $1.47 \pm 0.02$
> Qwen3-235B: $1.57 \pm 0.06$
> Qwen3-235B-Thinking: $1.85 \pm 0.03$
> GPT-OSS-120B: $2.08 \pm 0.05$
> Claude-4-Sonnet: $2.58 \pm 0.05$
>
> These analyses show that performance is consistent across runs and that there is a clear trend in performance between models, which aligns with our initial reported results. We plan to update all results for Biomni ablations (planner in Table 2, critics in Table 3, and retriever in Table 4) with three runs; however, a single run for the open-source models, Qwen3-235B-Thinking and GPT-OSS-120B, takes ~9 hours and ~6 hours, while serving the LLM on 8 and 4 GPUs, respectively. We plan to finish them in the upcoming days and update the tables once we finish all the runs. We appreciate your understanding and will update the status here.
>
> In response to the diversity of tasks in the benchmark, we have now defined six task categories: heterogeneity analysis, condition/treatment analysis, pathway analysis, key gene analysis, cellular functioning analysis, and cell-cell communication analysis. We provide the task distribution and correctness scores in Appendix C.3 and D.3. Upon close inspection, Claude-4-Sonnet consistently outperforms its open-source counterparts, particularly in condition/treatment analysis, heterogeneity analysis, and cell-cell communication analysis, achieving performance up to $\sim2\times$ that of open-source LLMs. The more challenging categories for all LLMs include key gene analysis and cellular functioning analysis, where Claude-4-Sonnet achieves $1.90$ and $2.16$ correctness scores, respectively, while other LLMs achieve lower scores, as low as $1.53$ in key gene analysis by Qwen3-235B-Thinking.
>
> Finally, we would like to emphasize that benchmarks in data-driven scientific discovery include a similar order of tasks as scHeurekaBench (102 in [3], 188 in [4], 198 in [5], 205 in [1]). This is consistent with benchmark sizes in other related domains, for instance, in Data Analysis (100 in [6], 257 in [7]) and ML (75 in [8], 13 in [9]). Further, [4] derives the questions from 12 datasets and research questions. This choice is partly because inference on these tasks can lead to expensive agentic rollouts, and constructing these benchmarks often involves manual validation.
>
> > W3. Since the evaluation hinges on LLM as a judge, it would be important to know how the LLM scores compare to human expert judgement. We do not know this right now.
>
> We appreciate the reviewer’s suggestion to compare LLM judge scores with human expert judgment. As this is a shared recommendation by all the reviewers, we provide the details of this new study in a separate official comment.
>
> > Q1. What are steps to ensure that the LLM/agent is doing the correct steps to obtain the correct insights? How do we avoid a hallucinated answer that happens to be correct?
>
> This is an interesting question. In open-ended scientific discovery, there is no single correct pathway (or straightforward sequence of steps) to derive findings. Indeed, we expect the agent to explore both the data and the accessible tools, and to query existing knowledge bases, before generating the final response. Therefore, the trajectories can contain repetitions and alternate decision paths that were not considered earlier, making it difficult to score them. Thus, we rely on the agent's final interpretation and answer.
>
> Hallucination is a significant open challenge within the LLM research, which aims to reduce, identify, and evaluate the extent and impact in LLM applications. To mitigate this problem, in our evaluation, we specifically look for atomic facts in the ground truth and agent-generated answers to verify if they are present and well supported by data-driven evidence.

---

> ### Author Response · Authors · 2025-11-21
> **Response to Reviewer eTC4 (3/3)**
>
> > Q2. How do we ensure that the LLM has not seen the data/publications in training, especially as new models come out? How does this impact our interpretation of results.
>
> We thank the reviewer for raising this interesting point that poses a challenge for creating benchmarks for evaluating LLM applications. Our framework yields questions that holistically evaluate scientific thinking, requiring the use of intrinsic skills to explore the data, as well as additional user-provided knowledge, existing scientific databases, and domain-specific tools. Poor performance on the benchmark would indicate these skills are missing.
>
> We also relied on recent publications from the past two years to minimize the likelihood that agents would depend solely on memorized knowledge when answering scHeurekaBench. We can use new scientific publications to evolve the benchmark by re-running it on newer publications alone. In response to the reviewer, we have now added this discussion to our Future work.
>
> > Q3. While the approach is different, the the focus on open-ended questions is not new [1]. It would be helpful to contextualize the motivation in the related work.
>
> We thank the reviewer for highlighting [4], which evaluates the intermediate decisions in data-driven analysis by posing them as multiple-choice questions. It uses crowdsourced efforts to collect annotations of intermediate analysis decisions and evaluates the open-endedness of scientific discovery by matching these decisions.
>
> First, [4] contains only multiple-choice questions, whereas our benchmark includes open-ended questions as well. Additionally, our framework evaluates the final data-driven response, accounting for the agent’s interpretation after its multi-step analysis. Furthermore, the evaluation in [4] is limited by the analysis steps in the crowdsourced efforts, which may not encompass all possible intermediate variables, decisions, and pathways. Meanwhile, our framework allows the agent to explore datasets and tools, make a non-linear sequence of decisions, and arrive at the final answer. It would be promising to combine the two evaluation frameworks to design a more holistic evaluation that assesses both intermediate steps and the final interpretation. We again thank the reviewer for highlighting this paper, and we have included it in our related work.
>
> **References**:
> [1] Mitchener et al. BixBench: a Comprehensive Benchmark for LLM-based Agents in Computational Biology. arXiv 2025.
> [2] Majumder et al. DiscoveryBench: Towards Data-Driven Discovery with Large Language Models. ICLR 2025.
> [3] Chen et al. ScienceAgentBench: Toward Rigorous Assessment of Language Agents for Data-Driven Scientific Discovery. ICLR 2025.
> [4] Gu et al. BLADE: Benchmarking Language Model Agents for Data-Driven Science. arXiv 2024.
> [5] Luo et al. Benchmarking AI scientists in omics data-driven biological research. arXiv 2025.
> [6] Sahu et al. InsightBench: Evaluating Business Analytics Agents Through Multi-Step Insight Generation. ICLR 2025.
> [7] Hu et al.  InfiAgent-DABench: Evaluating Agents on Data Analysis Tasks. ICML 2024.
> [8] Chan et al. MLE-bench: Evaluating Machine Learning Agents on Machine Learning Engineering. ICLR 2025.
> [9] Huang et al. MLAgentBench: Evaluating Language Agents on Machine Learning Experimentation. ICML 2024.

---

> > ### Author Response · Authors · 2025-12-01
> > **Follow-up response to Reviewer eTC4**
> >
> > In continuation of our response to W2, where we initially reported the averaged performance across three runs for a subset of planner LLMs in the agent, we have now updated our manuscript to include the average performance of agents across three runs for all other ablations. These analyses further show that performance and trends are consistent with our initial reported results.

---

### Official Review · Reviewer_3LvA · 2025-10-31

**Soundness:** 3
**Presentation:** 2
**Contribution:** 2
**Rating:** 4
**Confidence:** 3

**Summary:**

HEUREKABENCH is introduced as a benchmarking framework to evaluate LLM-based agents as co-scientists by unifying prior isolated subtasks and formalizing the task as triplets (D, Q, A) grounded in real experimental datasets, open-ended research questions, and scientifically validated answers. Using a multi-LLM, human-supervised pipeline over publications, datasets, and code, the authors instantiate the framework in single-cell biology as sc-HEUREKABENCH with 50 open-ended and 50 multiple-choice questions derived from validated insights. They further create sc-HEUREKABENCH-ToolUsage to assess domain-specific tool use and present experiments on the insight-construction modules and agent ablations (e.g., removing the retriever reduces correctness on TU).

**Strengths:**

1. Questions are grounded in real papers, data, and code, and require multi-step analysis and evidence-based reasoning rather than pure retrieval or recall—matching the “co-scientist” setting.
2. The benchmark compares multiple single-cell agents under a common setup and systematically ablates planner/critic/retriever to quantify their impact and provide design takeaways.

**Weaknesses:**

1. The evaluation relies on LLM-as-judge with an atomic-facts rubric (GPT-4o), and the manuscript does not report human adjudication or agreement statistics—leaving uncertainty and potential bias.
2. MCQ choices are LLM-generated; the authors acknowledge some “incorrect” options can appear scientifically plausible, so they also report precision/recall—indicating limited answer uniqueness that may complicate assessment.
3. Methodological details about the agent/tooling side remain sparse: the paper notes a retriever that pre-selects tools and that Biomni “contains more domain-specific tools and databases,” but it does not enumerate the tool inventory or selection criteria in detail.
4. Manual “minor code edits” (loading data, mapping gene IDs, renaming variables) are routinely applied to make verification run, which weakens the claim of end-to-end automation.

**Questions:**

1. How many tool categories are actually involved across the environment and TU tasks (beyond the named SCENIC, CellPhoneDB, CellChat, and NMF)?
2. The retriever that selects tools before planning seems necessary—does it implement a coarse-to-fine (e.g., candidate generation + re-ranking) procedure akin to RAG pipelines?
3. Since overall system results are reported only on sc-HEUREKABENCH-Lite (≤750 MB; 22/50 OEQs and 18/50 MCQs) due to cost/stability issues, how representative is this subset and how well does its distribution align with the full benchmark?
4. Figure 1(c) illustrates the question-solving stage; could the CellAgent-style planner–executor–evaluator pipeline implement this directly, or does your setup introduce additional mechanisms beyond such frameworks—for example, a retriever that pre-selects relevant tools, software, and databases before planning?
5. The pipeline produces 10 candidate insights per paper—what motivates this fixed number, and is it sufficient to capture the diversity of findings within each study?

---

> ### Author Response · Authors · 2025-11-21
> **Response to Reviewer 3LvA (1/2)**
>
> We thank the reviewer for their suggestions and valuable comments. We are delighted to see that the reviewer agrees that our benchmark questions match the co-scientist setting and found that our benchmarking of multiple single-cell agents with ablations of different agent components can provide key design takeaways. Below, we address the reviewer’s questions (Weaknesses and Questions are abbreviated as W and Q):
>
> > W1. The evaluation relies on LLM-as-judge with an atomic-facts rubric (GPT-4o), and the manuscript does not report human adjudication or agreement statistics—leaving uncertainty and potential bias.
>
> We thank the reviewer for their suggestion to include LLM judge and human agreement statistics to reduce uncertainty in the evaluation method. As this is a shared recommendation by all the reviewers, we provide the details of these new experiments in a separate official comment at the beginning of our response
>
> > W2. MCQ choices are LLM-generated; the authors acknowledge some “incorrect” options can appear scientifically plausible, so they also report precision/recall—indicating limited answer uniqueness that may complicate assessment.
>
> This is an interesting discussion. We would like to draw attention to the fact that incorrect options in MCQs should be plausible so that it is difficult for a co-scientist to solve the question by naive elimination; instead, these options should encourage the agent to use scientific tools and the provided experimental data to answer. Additionally, since the incorrect options are LLM-generated, some of them could actually be correct but are not marked as such by the LLM during the multiple-choice question generation stage. Therefore, we include precision and recall metrics to offer a different perspective on co-scientist performance.
>
> > W3. Methodological details about the agent/tooling side remain sparse: the paper notes a retriever that pre-selects tools and that Biomni “contains more domain-specific tools and databases,” but it does not enumerate the tool inventory or selection criteria in detail.
>
> We appreciate the reviewer’s comment. Biomni contains $105$ software packages, $59$ databases, and $150$ specialized biomedical tools, including genomics, cell biology (e.g.,`annotate_celltype_scRNA`, `create_scvi_embeddings_scRNA`, `gene_set_enrichment_analysis`, etc.). Additional details about the Biomni can be found in [1]. However, listing all these tools with their descriptions would not be feasible, since our work focuses on developing a benchmark-creation and evaluation framework for co-scientists in HeurekaBench. We have now added additional details about the Biomni agent design in Section 4.2 as suggested by the reviewer.
>
> > W4. Manual “minor code edits” (loading data, mapping gene IDs, renaming variables) are routinely applied to make verification run, which weakens the claim of end-to-end automation.
>
> We would like to clarify that our HeurekaBench framework comprises automated and human verification stages to ensure that questions are grounded in scientific literature and experimental data. Therefore, we denote our framework as semi-automatic. In particular, during the insight generation stage, all code-related modules (CodeDescriber, CodeMatcher, and CodeGenerator) are fully automated, followed by a human validation step that requires only minor code edits.
>
> > Q1. How many tool categories are actually involved across the environment and TU tasks (beyond the named SCENIC, CellPhoneDB, CellChat, and NMF)?
>
> The tools involved across the environment and TU tasks typically fall under the genomics and cell biology categories. The evaluated co-scientists can use any tools which they have access to within their defined environment. Our questions do not impose instructions or restrictions on the tools to use, which encourages the co-scientist to use tools that they identify as relevant. That said, in response to the reviewer’s question, we have identified some distinct tools that are regularly used, in addition to the named ones: `query_ensembl` and `query_reactome` (to interact with Ensembl and pathway databases), `annotate_celltype_scRNA` (cell type annotation with cell types based on gene markers), `annotate_celltype_with_panhumanpy` (cell type annotation of single-cell RNA-seq data using Panhuman Azimuth Neural Network), `gene_set_enrichment_analysis` (for gene set enrichment and pathway analysis), `create_scvi_embeddings_scRNA` (to create scVI embeddings for single cell integration tasks) along with all functions available from python packages (for example, scanpy, numpy, pandas).

---

> ### Author Response · Authors · 2025-11-21
> **Response to Reviewer 3LvA (2/2)**
>
> > Q2. The retriever that selects tools before planning seems necessary—does it implement a coarse-to-fine (e.g., candidate generation + re-ranking) procedure akin to RAG pipelines?
>
> We appreciate the reviewer’s question. However, we want to draw attention to the fact that we benchmark existing AI co-scientists by querying the agents with questions from scHeurekaBench and evaluating their responses. We do not incorporate any additional modules into the agents. The retriever is not introduced as part of our benchmark, but rather defined within the Biomni agent. The retriever is designed as an LLM that is prompted with all tools, software, and databases, along with their descriptions, and tasked with retrieving as many task-relevant tools as possible. It doesn’t involve a coarse-to-fine retrieval [1].
>
> > Q3. Since overall system results are reported only on sc-HEUREKABENCH-Lite ..., how representative is this subset and how well does its distribution align with the full benchmark?
>
> We thank the reviewer for this question. We have categorized our questions into six categories for both the full and lite versions of the benchmark: heterogeneity analysis, condition/treatment analysis, pathway analysis, key gene analysis, cellular functioning analysis, and cell-cell communication analysis. We observe that the task distribution is similar in both these versions. We have now added this distribution, along with task category definitions, to Appendix C.3 and Figure 4 in the revised manuscript.
>
> In addition, we compute the mean scores of three planner LLMs within the Biomni agent for each task category. The overall trend across both full and lite versions of the benchmark is consistent. The closed-source LLM consistently outperforms its open-source counterparts, noticeably in condition/treatment analysis and heterogeneity analysis in both benchmark versions. Further, within each task category, the rankings of the three LLMs based on the assigned correctness score remain identical. In response to the reviewer’s question, we have now included bar plots of correctness scores for each category in Figure 5 in the revised manuscript.
>
> > Q4. Figure 1(c) illustrates the question-solving stage; could the CellAgent-style planner–executor–evaluator pipeline implement this directly, or does your setup introduce additional mechanisms beyond such frameworks—for example, a retriever that pre-selects relevant tools, software, and databases before planning?
>
> We appreciate the reviewer’s question. We do not introduce additional mechanisms in our evaluation framework (Figure 1(c)). However, the CellAgent [2] codebase is not available online. Additionally, it is difficult to evaluate the agent for open-ended questions since in the CellAgent [2], the Evaluator must be defined with a suitable evaluation criterion so that it can select the optimal code sequences for a given task. Such an evaluator is challenging to design for open-ended scientific discovery, and it is unfair to use our LLM judges within the evaluator, as it requires access to the ground-truth answer to the open-ended question.
>
> > Q5. The pipeline produces 10 candidate insights per paper—what motivates this fixed number, and is it sufficient to capture the diversity of findings within each study?
>
> In our first attempt at generating insights, we extracted 5 and 10 insights (the range of high-level section headings in considered journal papers) using our InsightExtractor module and analyzed them. We observed that 10 extracted insights contained insights present in those extracted with five insights, along with additional and new insights. But it also included a few redundant (generic, very high-level, and typically in the last insight) and repetitive insights that were filtered during insight validation, indicating that we would not benefit from including additional insights. We provide some examples of redundant and repetitive insights below:
>
>   - Redundant (Generic):
>     - The study presents a comprehensive human skeletal muscle aging atlas, providing a valuable resource for studying muscle aging across species.
>     - The study provides a foundational framework for understanding the zonation of absorption across the mammalian small intestine, with implications for nutrient metabolism and disease.
>
>   - Repetitive:
>     - The caudal ganglionic eminence (CGE) is identified as the primary source of the postnatal EC stream, contributing to the continuous supply of interneurons.
>     - The CGE remains a hotspot for progenitor proliferation postnatally, contributing to the ongoing supply of neurons to the EC stream.
>
> We thank the reviewer again for their useful feedback. Note that all the changes in the revised manuscript have been highlighted in blue for clarity.
>
> **References**:
> [1] Huang et al. Biomni: A General-Purpose Biomedical AI Agent. biorXiv 2025.
> [2] Xiao et al. CellAgent: An LLM-driven Multi-Agent Framework for Automated Single-cell Data Analysis. arXiv 2024.

---

### Official Review · Reviewer_YHMR · 2025-11-01

**Soundness:** 3
**Presentation:** 3
**Contribution:** 3
**Rating:** 6
**Confidence:** 3

**Summary:**

This paper introduces HeurekaBench, a framework for creating benchmarks that evaluate LLM-based agents as AI co-scientists. The framework uses a multi-LLM pipeline to extract scientific insights from published papers and their code repositories, validates these insights through code execution, and generates open-ended and multiple-choice questions. The authors instantiate this framework in single-cell biology (sc-HeurekaBench) with 50 OEQs and 50 MCQs across 41 validated insights from 13 papers. They benchmark existing single-cell biology agents and analyze design choices including planner models, critic modules, and retrievers.

**Strengths:**

- The paper addresses an important gap in evaluating AI agents for scientific discovery by moving beyond simple factual recall or single-step computation tasks to assess genuine exploratory, multi-step data-driven reasoning capabilities.

- The semi-automated pipeline with human verification for insight validation is well-designed, ensuring that benchmark questions are grounded in reproducible scientific findings rather than relying solely on LLM generation capabilities.

- The experimental analysis provides valuable insights into agent design choices, demonstrating that critic modules can improve open-source model performance by up to 22% and that end-critics are more effective than plan-critics for certain models.

**Weaknesses:**

- The benchmark's scope is limited to only 13 papers and 50 questions in single-cell biology, raising concerns about generalizability and whether this sample size is sufficient to robustly evaluate agent capabilities across the diversity of real scientific discovery scenarios.

- The evaluation relies heavily on GPT-4o as an LLM judge for OEQs, which introduces potential biases and may favor agents using similar models, yet the paper provides limited analysis of inter-rater reliability or validation against human expert judgments.

**Questions:**

- As mentioned above, how sensitive are the evaluation results to the choice of LLM judge, and have the authors validated the scores against human expert assessments to ensure the rubric captures scientific correctness?

- What is the distribution of question difficulty across the benchmark, and could the authors provide more analysis on which types of insights or biological phenomena are most challenging for current agents to handle correctly?

---

> ### Author Response · Authors · 2025-11-21
> **Response to Reviewer YHMR (1/2)**
>
> We thank the reviewer for their valuable feedback and supportive review. We are happy to see the reviewer’s positive assessment of our framework for testing the genuine, exploratory, multi-step, data-driven reasoning capabilities of AI agents. Further, we are pleased that the reviewer finds our semi-automated pipeline with human verification for insight validation as well-designed to ground benchmark questions in reproducible scientific findings and that our experimental analysis provides valuable insights into agent design choices. Below, we address the reviewer’s questions (Weaknesses and Questions are abbreviated as W and Q):
>
> > W1. The benchmark's scope is limited to only 13 papers and 50 questions in single-cell biology, raising concerns about generalizability and whether this sample size is sufficient to robustly evaluate agent capabilities across the diversity of real scientific discovery scenarios.
>
> We thank the reviewer for the comment. The papers we consider cover diverse topics, including developmental biology, immune and infection studies, neurobiology, and comprehensive atlas studies. We selected all papers published over the last two years in two highly reputable journals (Nature and Cell), with paired data and code repositories, yielding 22 papers. Of these papers, 13 passed the validation processes of our benchmark, which ensures strong scientific grounding.
>
> In terms of task diversity, our benchmark covers six categories: heterogeneity analysis, condition/treatment analysis, pathway analysis, key gene analysis, cellular functioning analysis, and cell-cell communication analysis. In response to the reviewer’s feedback, we have now included (1) a new subsection C.2 in the appendix that discusses details about publications included in our benchmark, and (2) a new subsection C.3 in the appendix that provides details about task categories covered in our benchmark, and the number of questions per category.
>
> Moreover, we would like to emphasize that benchmarks for agentic systems in data-driven scientific discovery include a similar order of tasks as scHeurekaBench (102 in [1], 188 in [2], 198 in [3], 205 in [4]). This is consistent with benchmark sizes in other related domains, for instance, in Data Analysis (100 in [5], 257 in [6]) and ML (75 in [7], 13 in [8]). Further, [2] contains tasks derived from 12 datasets. This choice is partly because inference on these tasks can lead to expensive agentic rollouts, and constructing these benchmarks often involves manual validation.
>
> > W2. The evaluation relies heavily on GPT-4o as an LLM judge for OEQs, which introduces potential biases and may favor agents using similar models, yet the paper provides limited analysis of inter-rater reliability or validation against human expert judgments.
>
> > Q1. As mentioned above, how sensitive are the evaluation results to the choice of LLM judge, and have the authors validated the scores against human expert assessments to ensure the rubric captures scientific correctness?
>
> We thank the reviewer for their suggestion regarding inter-rater reliability and a study on alignment with human expert judgements, both of which strengthen our work. As this is a shared recommendation by all the reviewers, we provide the details of these new experiments in a separate official comment at the beginning of our response.

---

> ### Author Response · Authors · 2025-11-21
> **Response to Reviewer YHMR (2/2)**
>
> > Q2. What is the distribution of question difficulty across the benchmark, and could the authors provide more analysis on which types of insights or biological phenomena are most challenging for current agents to handle correctly?
>
> We appreciate the reviewer's question. We define the six task categories (in response to W1) and obtain the correctness scores of three LLMs in the Biomni agent for each category. We observe that Claude-4-Sonnet consistently outperforms its open-source counterparts, particularly in condition/treatment analysis, heterogeneity analysis, and cell-cell communication analysis, with Claude-4-Sonnet achieving performance up to $\sim2\times$ that of open-source LLMs. The more challenging categories for all LLMs include key gene analysis and cellular functioning analysis, where Claude-4-Sonnet achieves $1.90$ and $2.16$ correctness scores, respectively, while other LLMs achieve lower scores, as low as $1.53$ in key gene analysis by Qwen3-235B-Thinking.
>
> In response to the reviewer’s question, we have now included bar plots of these scores within each category, to show the difficulty across the benchmark, in Appendix D.3 and Figure 5.
>
> We thank the reviewer again for their useful feedback. Note that all the changes in the revised manuscript have been highlighted in blue for clarity.
>
>
>
> **References**:
> [1] Chen et al. ScienceAgentBench: Toward Rigorous Assessment of Language Agents for Data-Driven Scientific Discovery. ICLR 2025.
> [2] Gu et al. BLADE: Benchmarking Language Model Agents for Data-Driven Science. arXiv 2024.
> [3] Luo et al. Benchmarking AI scientists in omics data-driven biological research. arXiv 2025.
> [4] Mitchener et al. BixBench: a Comprehensive Benchmark for LLM-based Agents in Computational Biology. arXiv 2025.
> [5] Sahu et al. InsightBench: Evaluating Business Analytics Agents Through Multi-Step Insight Generation. ICLR 2025.
> [6] Hu et al.  InfiAgent-DABench: Evaluating Agents on Data Analysis Tasks. ICML 2024.
> [7] Chan et al. MLE-bench: Evaluating Machine Learning Agents on Machine Learning Engineering. ICLR 2025.
> [8] Huang et al. MLAgentBench: Evaluating Language Agents on Machine Learning Experimentation. ICML 2024.

---

### Official Review · Reviewer_jtc1 · 2025-11-01

**Soundness:** 2
**Presentation:** 3
**Contribution:** 2
**Rating:** 6
**Confidence:** 3

**Summary:**

This paper proposes HeurekaBench, a framework for creating benchmarks to evaluate LLM-based agents acting as AI co-scientists in data-driven, open-ended scientific discovery. Rather than relying on static knowledge retrieval or narrow computational tasks, they build benchmark  based on exploratory research questions that require multi-step analysis of real experimental datasets. They provide a semi-automated pipeline that leverages multiple LLMs to extract and validate scientific insights from published studies and their code repositories, grounding the benchmark in reproducible findings. The framework generates both open-ended questions to assess exploratory analysis and multiple-choice questions for rapid evaluation. Lastly, they instantiate the framework in single-cell biology as sc-HeurekaBench and conduct empirical experiments to evaluate state-of-the-art biological agents, analyzing the impact of key agent components like planners and critics.Their main contribution is proposing a framework for creating benchmarks to evaluate LLM-based agents acting as AI co-scientists, representing a paradigm shift toward evaluating the open-ended, data-driven reasoning required for AI co-scientists .

**Strengths:**

This paper proposes a novel framework for building benchmarks to evaluate LLM-based agents acting as AI co-scientists, effectively addressing the critical gap in evaluating open-ended, data-driven scientific discovery agent. Moreover, the paper is of relatively high quality, providing a comprehensive and clear exposition of its semi-automated pipeline and evaluation methodology, advancing beyond narrow task-solving, and demonstrating significant practical value of a critic module for the development of autonomous scientific agents.

**Weaknesses:**

The experiments in this paper are insufficient, as they lack comparative analysis with other benchmark construction methodologies[^1]. Without such comparisons, the work fails to fully demonstrate its superiority. Additionally, this study does not provide experimental analysis on the role of large language models as evaluators, such as comparing their performance with expert assessments to validate the reliability of this evaluation approach.

[^1]: Erpai Luo, Jinmeng Jia, Yifan Xiong, Xiangyu Li, Xiaobo Guo, Baoqi Yu, Lei Wei, and Xuegong Zhang. Benchmarking ai scientists in omics data-driven biological research. arXiv preprint arXiv:2505.08341, 2025.

**Questions:**

1.  The paper demonstrates the utility of HeurekaBench for evaluating agents, but how does the benchmark construction methodology itself compare to existing approaches, such as BaisBench , in terms of efficiency, insight quality, or scientific grounding?
2.  The study employs an LLM-as-a-judge for evaluating open-ended responses. To validate this approach, what is the correlation between the LLM's ratings and those from human domain experts, and have you analyzed specific cases where they disagree?

Suggestions:

1.  Include a comparative analysis with at least one other recent benchmark construction method (e.g., BaisBench) to more concretely position HeurekaBench's advantages and limitations within the field.
2.  Strengthen the validation of the evaluation method by conducting a small-scale study comparing the LLM-judge's scores with expert human assessments.

---

> ### Author Response · Authors · 2025-11-21
> **Response to Reviewer jtc1**
>
> We thank the reviewer for the valuable feedback and positive evaluation of our work. We appreciate the reviewer's positive comments regarding our paper quality, comprehensive exposition of the pipeline and evaluation. We are delighted that the reviewer finds that our framework has significant practical value for the development of autonomous agents and that it represents a paradigm shift toward evaluating the open-ended, data-driven reasoning required for AI co-scientists. Below, we address the reviewer’s questions (Weaknesses, Questions, and Suggestions are abbreviated as W, Q, and S):
>
> > W1. The paper demonstrates the utility of HeurekaBench for evaluating agents, but how does the benchmark construction methodology itself compare to existing approaches, such as BaisBench, in terms of efficiency, insight quality, or scientific grounding?
>
> > S1. Include a comparative analysis with at least one other recent benchmark construction method (e.g., BaisBench) to more concretely position HeurekaBench's advantages and limitations within the field.
>
> We thank the reviewer for the valuable suggestion about the comparative analysis with the BaisBench to position our work better. BaisBench utilizes a single LLM to create questions and answers from scientific publications. However, BaisBench question-answer pairs (1) have not undergone any validation, and (2) are not grounded in the experimental data but directly derived from publications. In contrast, our HeurekaBench framework uses scientific publications and corresponding code repositories to create question-answer pairs that (1) are generated from manually validated insights from scientific publications through code execution, and (2) are grounded in corresponding experimental data.  This ensures rigour at both the insight and question-generation stages, thereby ensuring the scientific grounding of the questions. Furthermore, compared to BaisBench, which focuses only on multiple-choice question-answer pairs, our framework also includes and evaluates open-ended questions.
>
> To evaluate the difficulty of (multiple-choice) questions (MCQs) from both benchmarks, we conducted a new evaluation where we used GPT-5 to answer MCQs from both BaisBench and HeurekaBench. In this setup, the LLM is given access only to questions, not to any experimental data. Hence, it tests the data-driven nature of the questions, i.e., whether they can be answered correctly based on knowledge recall. GPT-5 answered 53.37% of questions accurately on BaisBench, compared to 34.69% on scHeurekaBench, indicating that our framework yields questions that are more difficult to answer with recall alone.
>
> In response to the reviewer’s feedback, we expanded the discussion of the comparison with BaisBench and included a quantitative comparison of the MCQs provided by both frameworks in Appendix C.5.
>
>
> > W2. The study employs an LLM-as-a-judge for evaluating open-ended responses. To validate this approach, what is the correlation between the LLM's ratings and those from human domain experts, and have you analyzed specific cases where they disagree?
>
> > S2. Strengthen the validation of the evaluation method by conducting a small-scale study comparing the LLM-judge's scores with expert human assessments.
>
> We thank the reviewer for their suggestion to conduct a study on the alignment of our evaluation with human expert judgements, which strengthens our work. As this is a shared recommendation by all the reviewers, we provide the details of these new experiments in a separate official comment at the beginning of our response.
>
>
>
> We thank the reviewer again for their useful feedback. Note that all the changes in the revised manuscript have been highlighted in blue for clarity.

---

> > ### Comment · Reviewer_jtc1 · 2025-11-27
> >
> > Thank you for the detailed rebuttal and for adding the new experiments and analyses in response to my comments.
> >
> > In particular, the comparison with BaisBench helps clarify the positioning of HeurekaBench. This strengthens the case that your benchmark more directly tests data-driven scientific reasoning.
> >
> > Likewise, the new validation of the LLM-as-a-judge setup is very helpful. The small-scale study comparing GPT-4o’s correctness scores with those of 11 human single-cell experts provides strong evidence that the rubric and GPT-4o judgements are well aligned with expert assessments in this domain. The additional inter-rater agreement analysis across multiple LLM judges, with consistent agent rankings and reasonably high correlations and κ values, further increases my confidence in the robustness of the evaluation protocol.
> >
> > Overall, these additions address my main concerns about comparative positioning against other benchmark-construction methodologies and about the reliability of using an LLM as a judge for open-ended scientific answers. The work remains focused on a single domain and a relatively small but carefully curated set of tasks, so there is still room for future extensions to other scientific areas. However, I now feel more confident that the proposed framework is both scientifically grounded and empirically well calibrated. I therefore keep my score of 6 and would be happy to see this work accepted and further developed in future iterations.

---

> > > ### Author Response · Authors · 2025-11-27
> > > **Response to Reviewer jtC1**
> > >
> > > We are delighted to learn that we addressed all your concerns, including the comparison to BaisBench to position the HeurekaBench and validation of the LLM-as-a-judge setup with human experts and multiple LLM judges.
> > >
> > > Thank you for your time in reviewing our work and for the encouragement to accept our work.

---

### Author Response · Authors · 2025-11-21
**Inter-rater reliability across multiple LLM-based judges**

We thank the reviewers for their suggestion to conduct an inter-rater agreement study across multiple LLMs as judges to demonstrate the reliability of our proposed evaluation method.

We select three different closed-source LLMs as judges. In particular, we choose GPT-4o, Claude-4.5-Sonnet, and Gemini-2.5-Pro. These two additional judges do not belong to the GPT series of LLMs and were not used in any experiments. In parallel, we select the three best-performing planner LLMs within the Biomni agent (as per Table 2), i.e., Claude-4-Sonnet, GPT-OSS-120B, and Qwen3-235B-Thinking.

First, we compare the ranks assigned by the judges as per the correctness score. We observe that all judges agree on the ranking of the three planner LLMs: closed-source Claude-4-Sonnet is better than the open-source LLMs, and among open-source alternatives, GPT-OSS-120B is better than Qwen3-235B-Thinking. This ranking alignment is now shown in a new Figure 3.

Second, for each planner LLM, we compute more fine-grained inter-rater agreement metrics comparing the scores assigned by GPT-4o and other judges. In particular, we report Spearman's rank correlation and unweighted Cohen's kappa ($\kappa$) with quadratic penalty, which are suitable for ordinal discrete ratings. We find that the mean correlation across three planner models between GPT-4o and Claude-4.5-Sonnet judges is 0.84, whereas it is 0.79 with Gemini-2.5-Pro. Similarly, the $\kappa$ is 0.81 and 0.71, respectively. This study indicates high agreement between different LLM judges. The detailed metric for alignment between GPT-4o and other judges for each planner model is provided below (and in Table 6 of the revised manuscript):

**Claude-4-Sonnet planner**
| LLM Judges        | Spearman's rank correlation | Cohen's kappa κ |
| ----------------- | --------------------------- | --------------- |
| Claude-4.5-Sonnet | 0.863                       | 0.778           |
| Gemini-2.5-Pro    | 0.800                       | 0.651           |

**GPT-OSS-120B planner**
| LLM Judges        | Spearman's rank correlation | Cohen's kappa κ |
| ----------------- | --------------------------- | --------------- |
| Claude-4.5-Sonnet | 0.859                       | 0.849           |
| Gemini-2.5-Pro    | 0.798                       | 0.746           |

**Qwen3-235B-Thinking planner**
| LLM Judges        | Spearman's rank correlation | Cohen's kappa κ |
| ----------------- | --------------------------- | --------------- |
| Claude-4.5-Sonnet | 0.793                       | 0.789           |
| Gemini-2.5-Pro    | 0.766                       | 0.726           |


In conclusion, within this recommendation, we used different LLMs as judges to compute agreement between their agent rankings and correlation statistics. These additional experiments are now included in Section 4.3.1 and Appendix D.4.

---

### Author Response · Authors · 2025-11-21
**Validation of LLM-judge’s scores with expert human assessments**

We appreciate the common recommendation to include a study comparing LLM-judge scores with expert human assessments to strengthen our paper. Consequently, we selected 25 open-ended answers from Biomni (GPT-OSS-120B) and asked 11 human experts to assign a score between 1 and 5 (inclusive) for each agent response relative to the ground truth. We define our human experts as Ph.D. students or post-doctoral researchers with at least 1 year of experience working with single-cell data, including analyzing and drawing conclusions. The experts are from four different universities and six different labs.

To aggregate the individual scores and account for the diversity in human preferences, we calculated the mode and median of the experts' responses for each open-ended question. We first evaluated the difference between the aggregated human scores and the correctness scores, checking how often the difference for each question was ≤1. The difference was within this threshold in 92% (23 out of 25 questions) and 96% (24 out of 25 questions) of cases, for mode and median, respectively, suggesting strong agreement between the expert and G-eval scores.

Furthermore, we calculated Spearman’s rank correlation between the aggregated expert scores and LLM-assigned correctness scores, yielding 0.93 for mode-based and 0.90 for median-based aggregation, indicating a strong association. Finally, we computed Cohen's Kappa scores with quadratic weighting and obtained 0.85 for both aggregation methods.

In conclusion, within this common suggestion, we conducted an alignment study between GPT-4o and human experts. We have now added these results to Section 4.3.1 of the main text.

---

### Meta-Review · Area_Chair_xLZe · 2026-01-05

**Summary:**

The main reviewer concerns centered on three points: (1) whether HeurekaBench is sufficiently well-positioned relative to existing benchmark construction approaches, (2) the reliability and potential bias of using LLMs as judges for open-ended scientific answers, and (3) the limited scope of the instantiation in a single domain with a relatively small number of tasks, raising questions about generalizability and statistical robustness.

At the same time, several reviewers acknowledged that framing evaluation around grounded, data-driven scientific insights is a meaningful and effective angle, which, while potentially hard to scale broadly, provides new perspectives on agent evaluation and yields actionable insights into agent design and optimization.

**Reviewer Concerns:**

1. Positioning and novelty of the benchmark framework: reviewers jtc1 and eTC4 questioned whether HeurekaBench is clearly differentiated from prior benchmarks such as BaisBench and BLADE, and whether it truly constitutes a reusable framework rather than a domain-specific pipeline. The rebuttal addressed this by providing a direct comparison with BaisBench, clarifying differences in data grounding, validation via code execution, and inclusion of open-ended questions. This concern is largely resolved.

2. Validity of LLM-as-a-judge evaluation: all reviewers raised concerns about bias and reliability of GPT-4o as a judge. The authors conducted substantial new studies: alignment with 11 human experts, inter-rater agreement across multiple closed-source LLM judges, and correlation and kappa analyses. These additions were explicitly acknowledged by reviewers jtc1 and YHMR as addressing their core concerns. This issue is convincingly resolved.

3. Scale, diversity, and generalizability: reviewers YHMR, 3LvA, and eTC4 noted the limited number of papers, questions, and the focus on single-cell biology, as well as reliance on a lite subset for some experiments. The authors clarified task diversity, provided category-level analyses, showed consistency between full and lite benchmarks, and contextualized benchmark size relative to prior work. While these responses mitigate the concern, generalization beyond the current domain remains an acknowledged limitation rather than fully resolved.

4. Methodological clarity and automation: reviewer 3LvA and eTC4 highlighted sparse details on tool inventories, retrievers, and the need for manual code edits. The rebuttal clarified the semi-automatic nature of the framework, expanded descriptions of agent tooling, and justified human-in-the-loop validation. These concerns are partially addressed but still indicate some engineering opacity.

**Reviewer Scores:**

1. Reviewer jtc1 was marginally positive and explicitly stated that the rebuttal addressed all main concerns; the score would likely remain unchanged but solidly supportive.
2. Reviewer YHMR was also marginally positive and had key concerns about evaluation validity and diversity addressed; the score would likely remain stable.
3. Reviewer 3LvA was marginally negative; although the LLM-judge validation concern was addressed, broader concerns about methodology and representativeness are only partially resolved, so a score increase is unlikely.
4. Reviewer eTC4 was marginally negative and skeptical about framework generality; despite additional experiments and clarifications, their fundamental concern about generalization likely remains.

---

### Decision · Program_Chairs · 2026-01-26

Accept (Poster)